# NEK7 couples SDHB to orchestrate respiratory chain electron transport homeostasis that impedes liver fibrosis

Zhenzhen Sun [1,2,3,4,7], Le Sun[1,3,4,7], Hu Hua[1,3,4], Ying Ren[1,3,4], Wenping Zhu[1,3,4], Xu Wang[2], Wei Gu[2], Songming Huang [1,3,4], Dandan Zhong[1,5], Ying Sun [1,5 ✉], Yue Zhang [1,3,4 ✉], Aihua Zhang [1,3,4,6 ✉] & Zhanjun Jia [1,2,3,4,5 ✉]

The mode of electron transport in mitochondrial respiratory chain determines whether it generates energy or more reactive oxygen species (ROS), a key for cellular adaptation to diverse oxygen environments. However, the understanding of the mechanisms remains incomplete. Here, we find that NIMA-related kinase 7 (NEK7), targeted to mitochondria by its signal peptides, binds to succinate dehydrogenase complex iron sulfur subunit B (SDHB), stabilizing the spatial conformation of complex II and promoting forward electron transport. Deficiency of NEK7 in hepatocytes induces reverse electron transport (RET) and inhibits mitochondrial respiration, thereby promoting ROS generation, triggering spontaneous liver fibrosis and aggravating $CCl_4$-induced liver fibrosis, which can be attenuated by RET inhibitors. More importantly, NEK7 overexpression effectively alleviates $CCl_4$- and choline-deficient, high-fat diet-induced liver fibrosis. Overall, these findings highlight the pivotal role of NEK7 in orchestrating complex II and electron transport, providing new insights into the regulation of mitochondrial homeostasis and potential fibrosis treatments.

Mitochondrial homeostasis determines the fate of cell[1–3]. Any abnormality in the mitochondrial electron transport chain, tricarboxylic acid cycle, fusion, fission, etc., can lead to a disturbance in mitochondrial homeostasis, including abnormal mitochondrial complex activity, disordered electron transport, abnormal morphological dynamics and mtDNA levels, unbalanced energy, and reactive oxygen species (ROS) generation[4]. These can cause cell senescence or death, and ultimately changing the fate of organs or even the entire body[1]. The stability of the mitochondrial complex and orderly transfer of electrons in the respiratory chain are particularly crucial for maintaining mitochondrial homeostasis[5], which is regulated by both mitochondrial and nuclear gene-encoded proteins[6]. The former usually constitutes an important subunit of the mitochondrial complex, whereas the latter regulates the mitochondrial complex by entering the mitochondria[7,8]. Abnormal activity of the mitochondrial complex disrupts the balance of related metabolites and the path of electron transport, such as reverse electron transport, leading to a substantial increase in ROS production[9–11]. This mechanism has been extensively studied and is involved in the pathogenesis and progression of various diseases[10,12,13].

[1]Nanjing Key Laboratory of Pediatrics, Children's Hospital of Nanjing Medical University, Guangzhou Road #72, Nanjing, P. R. of China. [2]Department of Endocrinology, Children's Hospital of Nanjing Medical University, Guangzhou Road #72, Nanjing, China. [3]Department of Nephrology, Children's Hospital of Nanjing Medical University, Guangzhou Road #72, Nanjing, P. R. of China. [4]Jiangsu Key Laboratory of Early Development and Chronic Diseases Prevention in Children, Nanjing Medical University, Nanjing, P. R. of China. [5]Jiangsu Key Laboratory of Geriatric Precision Medicine and Aging Intervention, Xuzhou Medical University, Xuzhou, Jiangsu, P. R. China. [6]State Key Laboratory of Reproductive Medicine and Offspring Health, Nanjing Medical University, Nanjing, Jiangsu, China. [7]These authors contributed equally: Zhenzhen Sun, Le Sun. ✉e-mail: yingsun@xzhmu.edu.cn; zyflora2006@hotmail.com; zhaihua@njmu.edu.cn; jiazj72@hotmail.com

NIMA-Related Kinase 7 (NEK7) was previously reported to be located in the nucleus and cytoplasm, and was initially found to affect spindle assembly, centrosome separation and microtubule dynamics, participating in the regulation of mitotic initiation and progression[14,15]. Later, another important function of NEK7 was discovered; under certain signals (e.g. $K^+$ efflux, $Ca^{2+}$ signaling, chloride efflux, and mitochondrial ROS), it can directly bind to NLRP3 and initiate the assembly of the NLRP3 inflammasome, which is essential for its activation[16,17]. We previously found that NEK7 may promote cell cycle progression by regulating cyclin proteins, especially cyclin B1, thereby alleviating acute liver injury caused by acetaminophen[18], which demonstrated another role of NEK7 independent of mitosis and NLRP3 inflammasome activation.

Through further analysis of the RNA-seq omics of liver tissues from *NEK7* hepatocyte-specific knockout homozygous mice, we found that genes affected by NEK7 deficiency were significantly enriched in the oxidative phosphorylation (OXPHOS) pathway, and were closely linked to genes related to the mitochondrial inner membrane and respiratory complex. Furthermore, we unexpectedly discovered that NEK7 could enter the mitochondria and change the spatial conformation of mitochondrial complex II by binding to succinate dehydrogenase complex iron sulfur subunit B (SDHB), promoting the stability of electron transport in the respiratory chain and reducing ROS generation. More importantly, NEK7 can significantly ameliorated chronic liver fibrosis caused by various factors.

In this work, we find that NEK7 is localized in the mitochondria and reveal its important role in the regulation of mitochondrial complex and electron transport, which not only provides new insights into the molecular mechanisms of mitochondrial homeostasis, but also identifies potential targets for both mitochondrial dysfunction-related fibrotic diseases and the development of novel therapeutic strategies.

## Results

### NEK7 is localized in the mitochondria of hepatocytes, mediated by its MTS

First, we analyzed the RNA-seq data (de novo generated) of the liver tissues from *NEK7* hepatocyte-knockout homozygous (Hom) mice and wild-type (Ctrl) mice (Fig. 1a). Kyoto encyclopedia of genes and genomes (KEGG) analysis showed that the genes affected ($\log_2$FC < 0, $Q$ value < 0.05) by hepatocyte NEK7 deletion were significantly enriched in the OXPHOS pathway (top-ranked). Furthermore, by GO-C analysis, we found that these genes enriched in the OXPHOS pathway were mainly clustered in the inner mitochondrial membrane and were associated with the mitochondrial complex as well as the respiratory chain. We speculated that NEK7 may enter the mitochondria and affect its function.

We performed co-localization staining of NEK7 and MitoTracker in mouse primary hepatocytes (Supplementary Fig. 1a), AML-12 cells (Supplementary Fig. 1b, c), and HepG2 cells (Fig.1b). Confocal imaging and co-localization analysis intuitively showed a high overlap of NEK7 and the mitochondria (Fig.1b and Supplementary Fig. 1a–c), which was verified in the liver tissues (including the mouse liver tissues and the human liver tissues, Supplementary Fig. 1g, h) and in the co-localization of NEK7 with TOM20 in HepG2 cells (Supplementary Fig. 1d, e). We also isolated mitochondrial and cytoplasmic fractions from the mouse primary hepatocytes, AML-12 cells, HepG2 cells, and human liver tissues, and detected NEK7 protein levels, respectively. We observed endogenous NEK7 was localized in the mitochondria at a level similar to that in the cytoplasm (Fig.1c, d and Supplementary Fig. 1f, i).

To explore how NEK7 enters the mitochondria, we analyzed the structure of the NEK7 protein and noticed that there are two amino acid sequences, which are very consistent with the features of mitochondria-targeted signal peptides (MTS)[19,20], located at residues 61–96 and 121–163 (Fig.1e). In detail, they are enriched in positively

charged residues, interspersed with hydrophobic residues, close to the α-helix, and highly conservative among various species (Supplementary Fig. 2). Next, we constructed *NEK7-flag* (NEK7 full-length) and *NEK7$^{\triangle MTS}$-flag* plasmids (lack the above two predicted MTS sequences) and transfected them into HepG2 cells. Through co-localization analysis, we observed that the exogenous full-length NEK7 were also localized in the mitochondria, with a high overlap coefficient (mean overlap coefficient; full-length NEK7, 0.747), which was significantly reduced when both the MTS sequences were deleted (mean overlap coefficient; NEK7$^{\triangle MTS}$, 0.581), whereas with no significant change in the distribution of NEK7 in the nucleus and cytoplasm (Fig.1f–h). Furthermore, we constructed fusion plasmids of each MTS sequence with EGFP, and detected the localization of EGFP in sub-cellular structures. We found that both peptides could specifically target EGFP into the mitochondria (mean overlap coefficient: MTS$^{61-96}$, 0.850; MTS$^{121-163}$, 0.848) and even had a higher efficiency in entering the mitochondria than the full-length NEK7 sequence (Fig.1i-k). Taken together, for the first time, we identified the localization of NEK7 in the mitochondria and verified the possible MTS sequences for the protein entering the mitochondria.

### NEK7 deficiency leads to mitochondrial dysfunction

To investigate the role of NEK7 in the mitochondria, we first analyzed mitochondrial morphology by transmission electron microscopy and detected mitochondrial oxidative respiration. The ultra-structural images of the livers from the *NEK7* hepatocyte-specific knockout mice displayed that NEK7 deficiency in hepatocytes led to obvious abnormalities in the morphology of mitochondria, including decreased number of cristae, mitochondrial swelling, and vacuolization (Fig. 2a and Supplementary Fig. 3a), indicating serious mitochondrial damage and dysfunction[21,22]. Additionally, we observed obvious endoplasmic reticulum edema and disordered structure in the liver tissue of Hom mice (compared with the wild-type [WT] mice) (Fig.2a), which may be secondary to the mitochondrial damage, resulting from the long-term complete loss of NEK7 in hepatocytes. By a seahorse assay, we found the oxygen consumption rate (OCR) was significantly reduced in the primary hepatocytes isolated from the heterozygous (Het) mice, accompanied by increased mitochondrial ROS (mtROS), compared to hepatocytes from the WT mice (Fig.2b–d and Supplementary Fig. 3b). The mitochondrial membrane potential was significantly increased in *NEK7*-knockdown liver tissues (from the Het mouse), as determined by TMRE (tetramethylrhodamine ethyl ester perchlorate) staining (Fig.2e), further indicating the important role of NEK7 in mitochondrial function.

Furthermore, in vitro, the HepG2 cells with inhibition of NEK7 exhibited similar edema and vacuolization of mitochondria (Fig.2f and Supplementary Fig. 3c) as the hepatocytes of the Hom mice. We observed autophagy lysosomes in NEK7-inhibited cells (Fig.2f), which may have been formed to degrade damaged mitochondria[23]. Additionally, loss of NEK7 significantly inhibited mitochondrial oxidative respiration. This was confirmed in diverse human hepatocyte cell lines (Fig.2g–j and Supplementary Fig. 3c, d). NEK7 deficiency also resulted in a significant increase in mitochondrial ROS (Fig.2k, l) and mitochondrial membrane potential (Fig.2m, n) both in the HepG2 and HepaRG cells measured by flow cytometry. All these results suggested that loss of NEK7 resulted in mitochondrial dysfunction. In summary, mitochondria-localized NEK7 is essential for maintaining mitochondrial function in hepatocytes, and NEK7 deficiency leads to disturbed mitochondrial homeostasis, including abnormal mitochondrial morphology and respiration, and increased mitochondrial membrane potential and ROS.

### NEK7 regulates complex II conformation and electron transport by binding to SDHB

To investigate the downstream molecule of NEK7 involved in the regulation of mitochondrial function, we purified mitochondrial

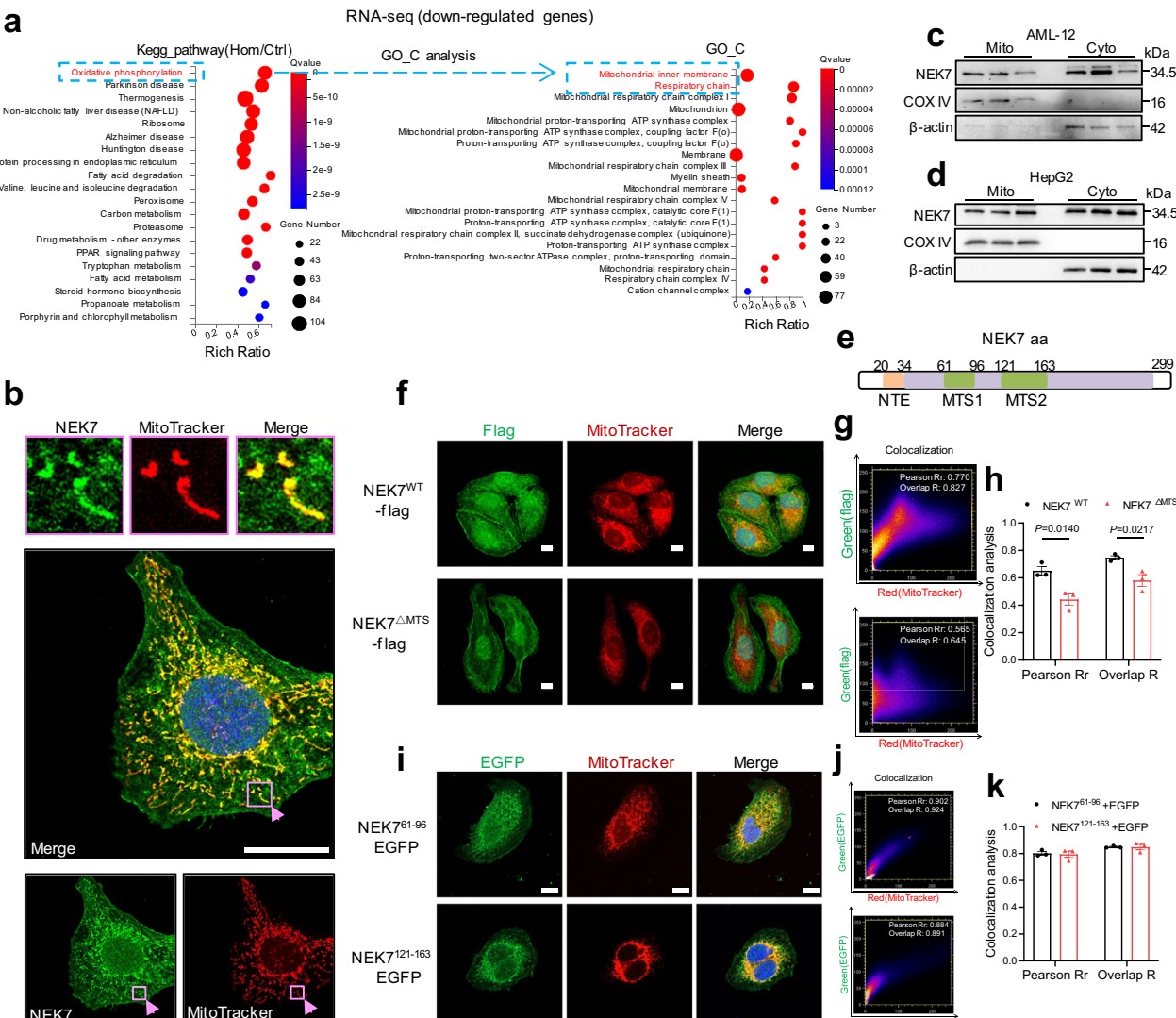

**Fig. 1 | NEK7 is localized in mitochondria, mediated by its MTS peptides. a** KEGG and GO-C analysis of the downregulated genes (compared to the WT mice livers) in the Hom mice livers. N = 3 mice/group. The Hom mouse (NEK7fl/fl -Alb-Cre + ) is the homozygous hepatocyte *NEK7* knockout mouse depicted in Fig. 4a. **b** Representative images acquired by laser confocal microscopy showing NEK7 localization (Green) in the mitochondria (Red, MitoTracker) of HepG2 cells. Scale bar, 20 μm. The enlarged images above show areas wherein NEK7 (green) co-localizes with mitochondria (red) in HepG2 cells (*n* = 3). **c,d**, Western blotting analysis showing the NEK7 protein levels in mitochondria and cytoplasm isolated from AML-12 and HepG2 cells (*n* = 3). Mito, mitochondria; Cyto, cytoplasm. **e** Distribution of the two MTS peptides (61–96 and 121–163) in NEK7 amino acid sequences. Refer to

Supplementary Fig. 2. **f–h** Representative images (**f**) and the co-localization analysis (**g, h**) showing the different distribution pattern of NEK7WT-flag (full-length NEK7) and NEK7ΔMTS-flag (the predicted MTS sequences are deleted) in the mitochondria of HepG2 cells. N = 3/group. Scale bar, 10 μm. Flag, green; MitoTracker, Red. **i–k** Representative images (**i**) and the co-localization analysis (**j, k**) showing the distribution pattern of EGFP in the HepG2 cells transfected with NEK761–96-EGFP or NEK7121-163-EGFP plasmids. N = 3 /group. Scale bar, 10 μm. EGFP, green; MitoTracker, Red. Data are presented as mean ± SE. Significant differences are analyzed using unpaired two-tailed *t*-test (**h**). Related data shown in Supplementary Fig. 1 and Supplementary Fig. 2. *N* values in (**b–d**, **f–k**) indicate biological independent replicates. Source data are provided as a Source Data file.

proteins from HepG2 cells and performed a co-immunoprecipitation assay using an NEK7 antibody, followed by mass spectrometry detection. Cluster analysis revealed that NEK7-related proteins in the metabolic module were mainly enriched for energy production and conversion (Fig. 3a). Considering the above results indicating the role of NEK7 in mitochondrial OXPHOS, we focused on proteins (including SDHB, TDIF2, and VATC1) that are NEK7-unique (compared with IgG) and related to the OXPHOS pathway, among which only SDHB is localized in the mitochondria and has an established link to mitochondrial respiration (Fig.3a). We verified the mass spectrometry results by CO-IP-western blotting in mouse primary hepatocytes, AML-12 cells, and HepG2 cells (Supplementary Fig. 4a and Fig.3b). Additionally, through co-localization analysis, we observed that NEK7 and

SDHB had a high overlap in the mitochondria of primary hepatocytes, AML-12, and HepG2 cells (Supplementary Fig. 4b–f and Fig. 3c, d).

Succinate dehydrogenase complex iron sulfur subunit B is a subunit of succinate dehydrogenase (also known as mitochondrial complex II), which is composed of four subunits (SDHA/SDHB/SDHC/SDHD) and cofactors (including FAD, three iron-sulfur clusters, and coenzyme Q)[24,25]. Succinate dehydrogenase is a key enzyme in the tricarboxylic acid cycle (involved in the oxidation of succinate to fumarate) and in the electron transport chain[25,26]. Numerous studies have uncovered that abnormal expression or mutation of SDHB affects mitochondrial complex II assembly, causes mitochondrial dysfunction, and is associated with the development of various tumors[27–29]. However, the regulatory mechanisms underlying the effects of SDHB

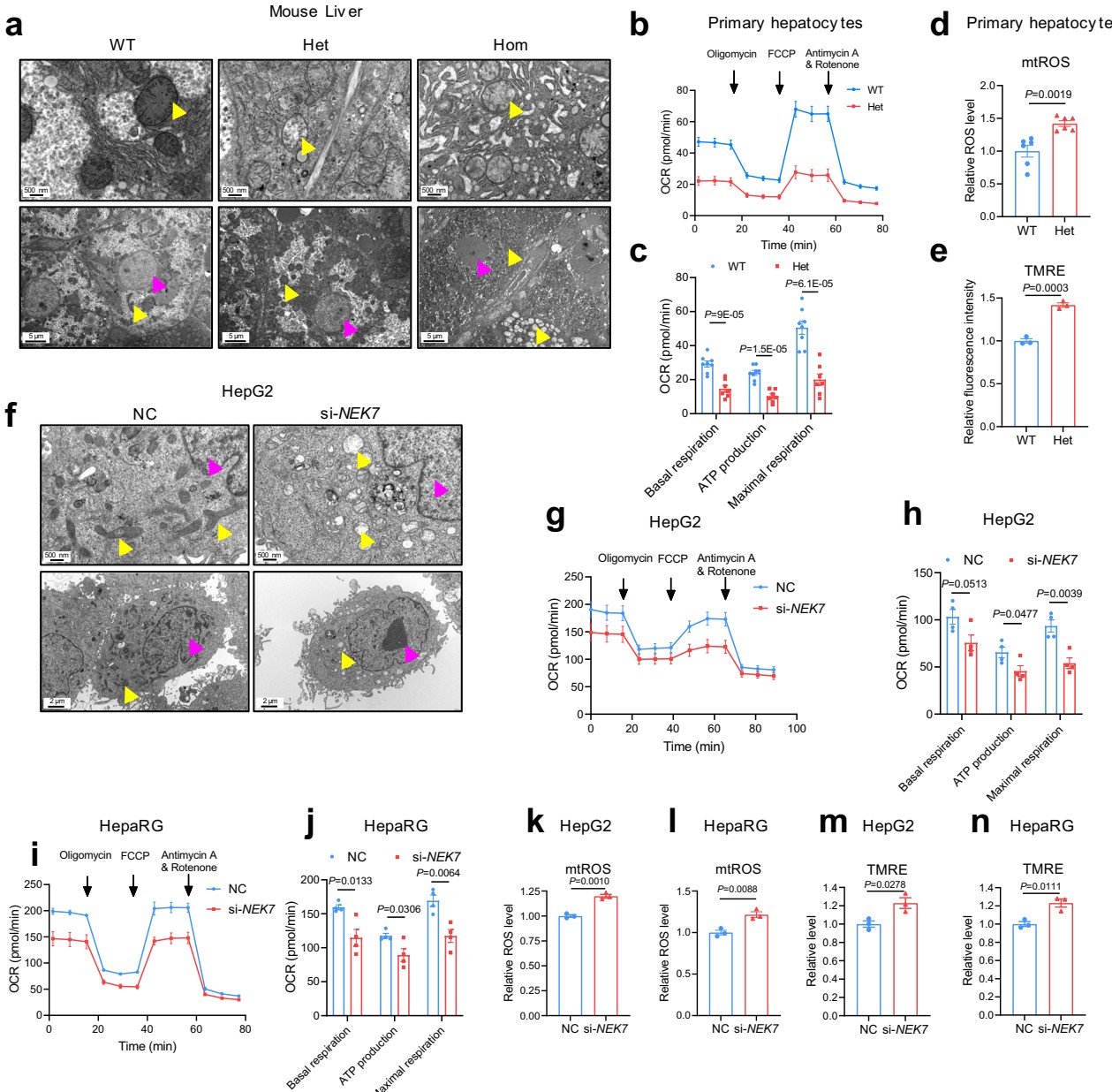

**Fig. 2 | NEK7 deficiency leads to mitochondrial dysfunction. a** Representative TEM images depicting the ultrastructure of the liver tissues from WT ($n = 3$ mice), Het (NEK7$^{fl/wt}$-Alb-Cre + , as depicted in Fig. 4a, $n = 5$ mice), and Hom mice (NEK7$^{fl/fl}$-Alb-Cre + , as depicted in Fig. 4a, $n = 3$ mice). Pink and yellow arrows point to the nucleus and mitochondria, respectively. Scale bar, 5 µm (below) or 500 nm (above). **b**, **c** Mitochondrial OCR of the primary hepatocyte (WT, $n = 8$; Het, $n = 7$), isolated from the WT and Het mice. **d** Flow cytometry detected mitochondrial ROS levels (MitoSox Red) in the primary hepatocytes (isolated from the WT and Het mice). $N = 6$/group. **e** Quantification of mitochondrial membrane potential determined by TMRE staining in the WT and Het mice livers. $N = 3$/group. **f** TEM acquired images depicting the ultrastructure of the HepG2 cells transfected with NC (negative control) or si-*NEK7* for 48 h. The pink arrows point to the nucleus, and the yellow arrows point to the mitochondria. Scale bar, 2 µm (below) or 500 nm (above). N = 3/

group. **g**, **h** Mitochondrial oxygen consumption rate (OCR) in HepG2 cells determined by seahorse assay, $n = 4$/group. The HepG2 cells are treated as in (**a**). **i**, **j** Mitochondrial OCR in HepaRG cells determined by seahorse assay, $n = 4$/group, the HepaRG cells are treated as in (**a**). **k**, **l** Flow cytometry detected mitochondrial reactive oxygen species (mtROS) levels (MitoSox Red) in the control and NEK7-inhibited hepatocytes (including HepG2 and HepaRG cells). $N = 3$/group. **m**, **n** Flow cytometry detected mitochondrial membrane potential (TMRE) in control and NEK7-inhibited hepatocytes (including HepG2 and HepaRG cells). $N = 3$/group. Data are presented as mean ± SE. Significant differences are analyzed using unpaired two-tailed *t*-test (**c**–**e**, **h**, **j**–**n**). Related data shown in Supplementary Fig.3. *N* values in (**b**–**n**) indicate biological independent replicates. Source data are provided as a Source Data file.

remain elusive. We found that neither over-expression nor low expression of NEK7 affected SDHB protein levels in HepG2 cells (Fig. 3e, f). Additionally, SDHB was not altered in the primary hepatocytes isolated from hepatocyte *NEK7* knockdown mice (the Het mice) or in the liver tissue of adult mice with hepatocyte *NEK7* knockdown induced by AAV8-TBG-cre (Fig. 3g, h). The complex II assembly and integrity were also unaffected whether NEK7 was overexpressed or

silenced in the HepG2 cells determined by Blue Native Page assay (Supplementary Fig. 4g, h).

Furthermore, molecular docking analysis (Fig. 3i, j) revealed direct binding between NEK7 and SDHB (including salt bridges and hydrogen bonds), and that leads to slight increase in the distance of electron transport within the SDH complex (measured after the molecular dynamics simulation). Microscale thermophoresis (MST) and surface

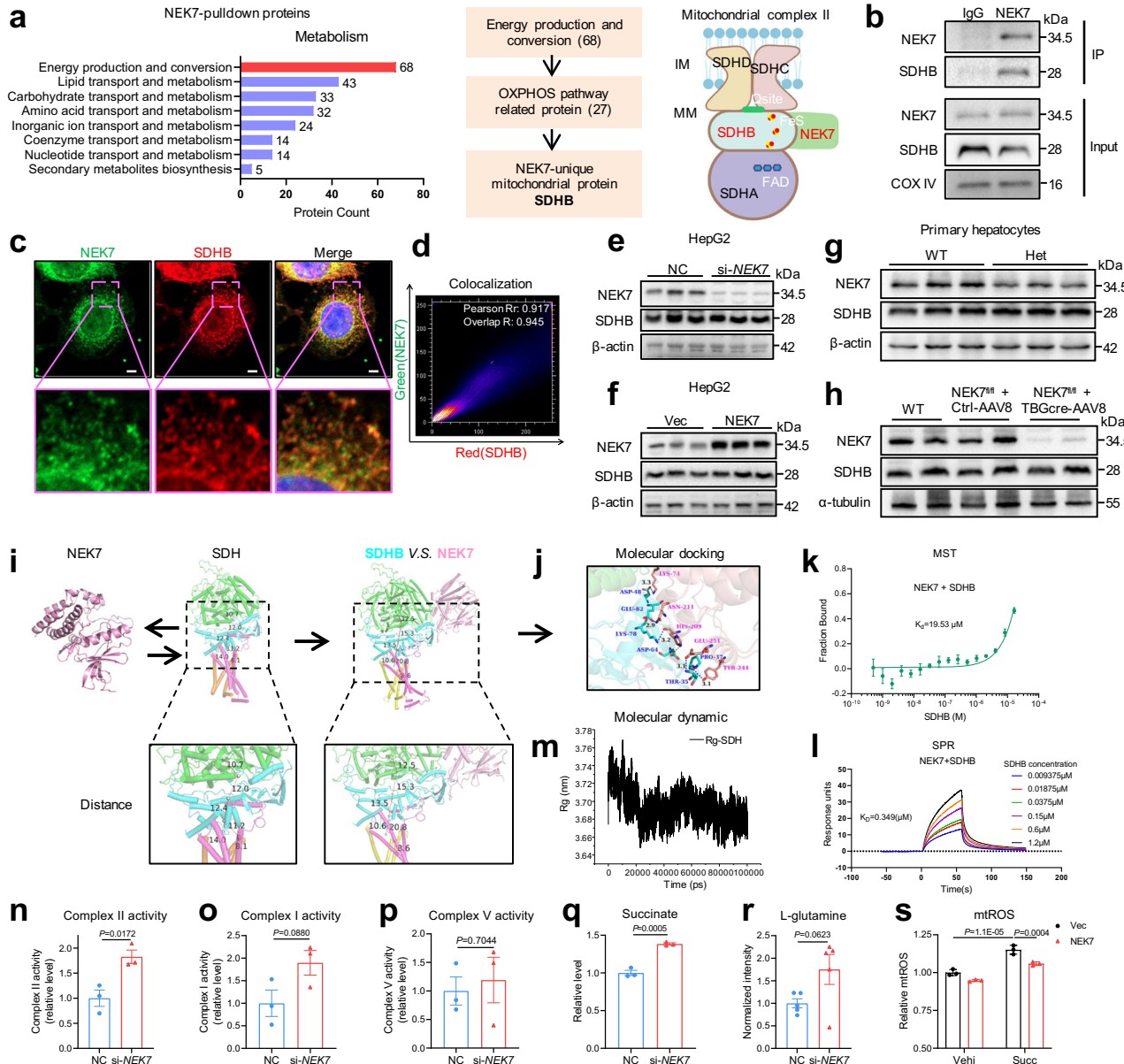

**Fig. 3 | NEK7 regulates complex II conformation and electron transport by binding to SDHB. a** NEK7-related mass spectrometry depicting the number of proteins enriched in diverse functional categories of metabolic modules (left). The diagram in the middle depicts that, of the 68 proteins (pulldown by NEK7 antibody) enriched in energy production and conversion, SDHB is the only mitochondrial protein related to the OXPHO pathway. The diagram on the right draws the possible interaction between NEK7, SDHB, and mitochondrial complex II. Created in BioRender. Sun, z. (https://BioRender.com/l2fzi06). **b** Co-immunoprecipitation analysis showing the interaction of NEK7 and SDHB determined by IgG and NEK7 antibody. N = 3/group. **c, d** Representative IF images (**c**) and the co-localization analysis (**d**) depict NEK7 and SDHB co-localization in the HepG2 cells. Scale bar, 5 μm. N = 3/group. **e,f**, Western blotting analysis of NEK7 and SDHB protein levels in the HepG2 cells transfected with si-*NEK7*/NC or NEK7/Vec (vector) plasmids for 48 h. N = 3/group. **g, h** Western blotting analysis of SDHB level in NEK7-depleted (from the Het/WT mice) primary hepatocytes (n = 3/group) and liver tissues (from the NEK7^fl/fl

mice injected with AAV8-TBGcre-EGFP for four weeks, n = 4 mice/group). **i–j** Molecular docking analysis of NEK7 and SDHB. **k** MicroScale Thermophoresis (MST) analysis of NEK7 and SDHB protein. **l** Surface Plasmon Resonance (SPR) analysis of NEK7 and SDHB protein (Supplementary Data 2). **m** Molecular dynamics simulation showed the gradually decreasing Rg (Radius of gyration) value of SDH complex after SDHB binding with NEK7 (Supplementary Data 3-6). **n–p** Activities of mitochondrial complex II, I, and V in the HepG2 cells transfected with si-*NEK7*/NC for 48 h. N = 3/group. **q, r** Level of Succinate (n = 3/group) and L-glutamine (n = 5/group) in the HepG2 cells transfected with si-*NEK7*/NC for 48 h. **s** Relative mitochondrial ROS level (MitoSox Red) detected by flow cytometry. The control (Vec) and NEK7-overexpressed HepG2 cells are treated with vehicle (Vehi) or Succinate (Succ, 1 mM) for 4 h. N = 3/group. Data are presented as mean ± SE. Significant differences are analyzed using unpaired two-tailed *t*-test (**n**–**r**), and a one-way ANOVA with Benjamini multiple comparisions (**s**). N values in (**b**–**g**, **n**–**s**) indicate biological independent replicates. Source data are provided as a Source Data file.

plasmon resonance (SPR) analysis also showed that NEK7 and SDHB protein had a good binding affinity (Fig. 3k, l), which was decreased significantly with the deletion of binding sites in NEK7 (the $K_D$ value increases from 0.349 μm to 4.750 μm) (Supplementary Fig. 4i), further demonstrating the possibility of direct interaction between them. Additionally, using molecular dynamics simulation[30,31], we found that

NEK7 altered the spatial conformation of SDH complex after binding to SDHB, which tended to be more stable, as evidenced by a decreased gyration radius (Fig. 3m).

Next, we determined the activity of the mitochondrial complex II. In HepG2 cells, NEK7 overexpression did not significantly affect the complex II activity (Supplementary Fig. 4j). However, surprisingly,

NEK7 deficiency led to a significant increase in the activity of mitochondrial complex II, and just an arising trend in the activity of mitochondrial complex I, whereas the mitochondrial complex V activity did not change (Fig.3n-p). Combined with the above seahorse results and increased membrane potential and mitochondrial ROS generation (Fig.2), we speculate that the unstable SDH complex, due to the absence of NEK7 bound to SDHB, leads to uncoupling between electron transport and OXPHOS in the mitochondria[32–35]. Suppressed generation of ATP, increased membrane potential and mtROS levels, along with elevated mitochondrial complex II activity are the important indicators of reverse electron transfer (RET)[11,32–35]. Thus, all the above evidence suggests there may prompt RET in the respiratory chain in hepatocytes with NEK7 knockdown. That is, the electrons from mitochondrial complex II cannot be transferred forward to generate energy, but are reversely transferred to FAD or mitochondrial complex I, resulting in the leakage of more electrons, which is an important mechanism that leads to a large amount of mitochondrial ROS generation and limited energy production[11,36].

To investigate why NEK7 deficiency causes RET, we detected the levels of related metabolites. We found that the level of succinate (also known as succinic acid) was significantly increased in the HepG2 cells with NEK7 knockdown (Fig.3q), a key metabolite that leads to the generation of mitochondrial ROS by inducing RET[10,35]. The level of L-glutamine was also increased (Fig.3r), which could enter the mitochondria to promote further production of succinate[37,38]. Here, increased succinate and L-glutamine levels in hepatocytes, whether directly or indirectly associated with NEK7 deficiency, may further enhance RET. Next, we treated the HepG2 cells with succinate, which induced the generation of mitochondrial ROS (Fig.3s), consistent with the results of previous studies. Succinate-induced mitochondrial ROS was significantly reduced when NEK7 was simultaneously overexpressed in the cells (Fig.3s). These results further suggest that the occurrence of RET[11,13] due to NEK7 deficiency probably originated from the sites of mitochondrial complex II.

Moreover, we explored the effect of SDHB on mitochondria. We found that inhibiting SDHB induced the generation of mitochondrial ROS, and increased mitochondrial membrane potential (Supplementary Fig.5a–c). However, the activity of mitochondrial complex II was significantly down-regulated (Supplementary Fig. 5d), suggesting that the loss of SDHB may lead to a reduction or abnormal assembly of mitochondrial complex II. This also indicates that SDHB is essential for the stability of mitochondrial complex II.

Taken together, we demonstrate the important role of NEK7 in maintaining the homeostasis of mitochondrial complex II and electron transfer, through binding to SDHB.

## Hepatocyte *NEK7* knockout mice develop spontaneous hepatic dysfunction and fibrosis

During the breeding of hepatocyte *NEK7* knockout mice (Fig. 4a, b), we unexpectedly found that the stress urine of 5-month-old Hom mice was more yellow than that of the WT mice (data not shown). After phenotype analysis, it was surprising that the livers of Hom mice were harder and sharper, and their serum was obviously more yellow than that of the WT mice (Fig. 4c). Additionally, in the Hom mice, liver function, indicated by the levels of alanine and aspartate aminotransferases (ALT and AST), decreased sharply, the levels of direct bilirubin (DBIL) in the serum were significantly increased, and liver pathological injury was more serious (Fig.4d–h and Supplementary Fig. 6a). The Het mice had mild liver dysfunction and structural abnormalities as shown by serum biochemical analysis and liver histopathological staining (Fig.4d–g and Supplementary Fig. 6a). Next, we measured the level of liver fibrosis and inflammation by various experiments. In the Het mice, we observed mild fiber deposition only limited in the portal areas of the liver, and inflammation increased slightly (Fig.4h–r and Supplementary Fig. 6b-l). In the Hom mice, the liver exhibited diffuse fibrosis with abundant fiber deposition, and the mRNA and protein levels of various fibrotic molecules were markedly increased (Fig.4h-r and Supplementary Fig. 6b-l). Moreover, the level of inflammation was also significantly increased, as shown by the mRNA levels of inflammatory cytokines, accompanied by a large number of inflammatory cell infiltration detected by CD45 staining (Fig.4h, o-q and Supplementary Fig. 6d, h). Additionally, we found that liver function and fibrosis in the Het mice further deteriorated at 10-month (Supplementary Fig. 7).

We also observed that *NEK7* knockout in the hepatocytes significantly induced proliferation and activation of hepatic stellate cells in the Hom mice, which were characterized by increased distribution of the cells and increased protein and mRNA levels of α-SMA (Fig. 4h, m, r). Based on the above results, we speculate that NEK7 deficiency-induced mitochondrial dysfunction and oxidative stress cause hepatocyte damage, which may promote the proliferation and activation of the hepatic stellate cells through paracrine effects and finally lead to pathological progression in the liver (Fig.4s). Furthermore, in vitro, we applied the medium of the HepG2 cells with inhibition of NEK7 expression to culture LX-2 cells (human hepatic stellate cell line), which significantly increased the levels of α-SMA and fibrosis-related genes (Fig. 4t-v), indicating activation of the LX-2 cells. Moreover, we observed similar results when LX-2 cells were cultured with the medium of SDHB-inhibited HepG2 cells (Fig. 4w–y and Supplementary Fig. 6m, n). Together, these results showed that *NEK7* knockout in hepatocytes progressed to severe liver dysfunction, pathological injury, and fibrosis, further indicating the important role of NEK7 in maintaining mitochondrial homeostasis in hepatocytes.

## Inhibiting RET alleviates the aggravation of CCl₄-induced liver fibrosis caused by hepatocyte *NEK7* knockdown

Next, we explored whether inhibition of RET or mitochondrial ROS could alleviate hepatocyte injury and the activation of hepatic stellate cells caused by NEK7 deletion. We used several RET inhibitors, including dimethyl malonate (DMM, target complex II)[39,40], rotenone (Rot, target complex I)[32,37], and metformin (Met, target complex I)[37], and the ROS scavenger Tempo[41,42]. We found that DMM, Rot, Met, and Tempo significantly reduced ROS levels resulting from NEK7 deletion in the HepG2 cells (Fig. 5a). Additionally, activation of the LX-2 cells induced by the medium of the HepG2 cells with NEK7 depletion was also significantly inhibited, as evidenced by decreased levels of α-SMA and other fibrosis-related proteins (Fig. 5b–d and Supplementary Fig. 8). DMM treatment alone did not significantly affect mitochondrial homeostasis in the HepG2 cells, as evidenced by no significant difference in the ROS level, mitochondrial membrane potential, and complex II activity between groups, as well as the level of α-SMA in the LX-2 cells with DMM-conditioned medium of HepG2 (Supplementary Fig. 9a-e). This suggested that the protective effect of DMM might depend on abnormal complex II activity caused by NEK7 deficiency.

In vivo, we used hepatocyte *NEK7* knockdown mice (the Het mice) and the corresponding WT mice as controls to induce liver fibrosis by intraperitoneal injection of CCl₄ for 6–7 consecutive weeks. We found that liver fibrosis in the Het mice was more severe than that in the WT mice, which was significantly attenuated when treated with DMM or Met, as well as inhibited proliferation and activation of the hepatic stellate cells (Figs.5e–j,5m–o). Lipid oxidative damage to the liver tissue was also significantly alleviated (Fig.5k, l). Furthermore, the aggravated damage of mitochondria in the livers by NEK7-knockdown were rescued by DMM as evidenced by TEM analysis (Supplementary Fig. 9f), suggesting that mitochondrial structural disorder may be secondary to the impaired activity of complex II. Additionally, in vitro, abnormal activity of complex II induced by CCl₄ could be restored by NEK7 overexpression (Supplementary Fig. 9g). This indicated that aggravation of CCl₄-induced liver fibrosis in the Het mice may be due

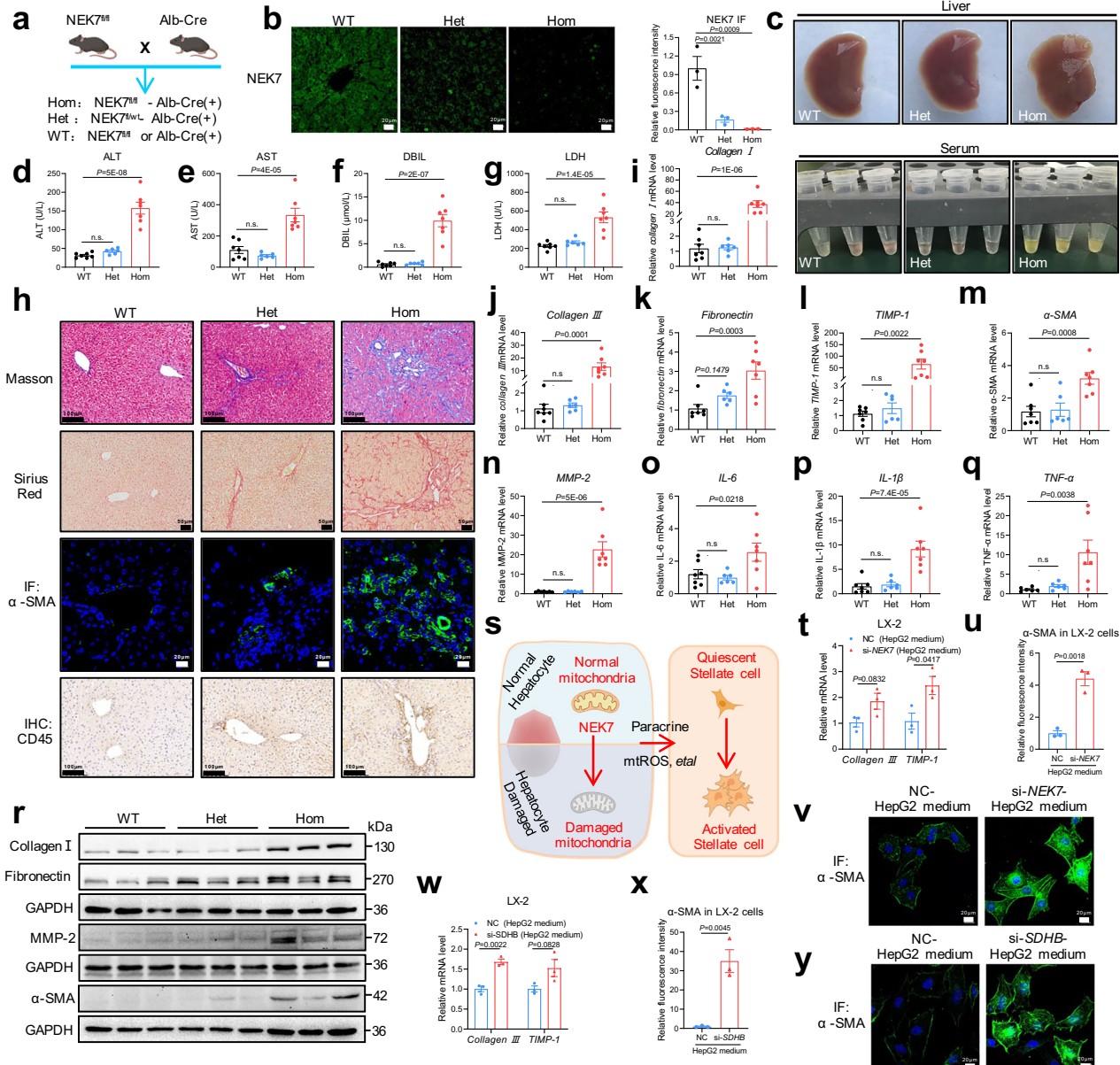

**Fig. 4 | Hepatocyte NEK7 knockout mice develop spontaneous hepatic dysfunction and fibrosis. a** Breeding scheme of the hepatocyte *NEK7* knockout mice. Created in BioRender. Sun, z. (https://BioRender.com/l2fzi06). **b** Representative images of NEK7 immunofluorescence (IF) staining in the mouse livers. N = 3/group. **c** Representative pictures of the livers (WT, *n* = 7; Het, *n* = 6; Hom, *n* = 7) and serum. **d–g** Levels of serum ALT, AST, LDH, and DBIL (WT, *n* = 7; Het, *n* = 6; Hom, n = 7). **h**, Representative images of Masson (Het, *n* = 6), Sirius Red (Het, *n* = 6), α-SMA IF (Het, *n* = 6), and CD45 IHC (Het, *n* = 5) staining, WT, n = 7; Hom, *n* = 7. The quantification in Supplementary Fig.6e-h. **i–q** Relative mRNA levels of *Collagen* I, *Collagen* III, *Fibronectin, TIMP-1, α-SMA, MMP-2, IL-6, IL-1β*, and *TNF-α*; WT, *n* = 7; Het, *n* = 6; Hom, *n* = 7. **r** Western blotting showing the expression of Collagen I, Fibronectin, α-SMA, and MMP-2 in the mouse livers, *n* = 6/group. The quantification in Supplementary Fig.6i–l. **s** The diagram depicts the speculation that NEK7 deficiency-induced mitochondria dysfunction and mtROS generation cause hepatocyte damage, which may promote proliferation and activation of the hepatic stellate cells through paracrine effects and results in the pathological progression in the liver. Created in BioRender. Sun, z. (https://BioRender.com/l2fzi06). **t** Relative mRNA levels of *Collagen* III and *TIMP-1* in the LX-2 cells, which are treated with the conditional medium of the HepG2 cells (transfected with si-*NEK7*/NC for 48 h) for 72 h. N = 3/group. **u, v** Quantification (**u**) and representative images (**v**) of α-SMA staining in the LX-2 cells with the same treatment as in (**t**). N = 3/group. **w** Relative mRNA levels of *Collagen* III and *TIMP-1* in the LX-2 cells, which are treated with the conditional medium of the HepG2 cells (transfected with si-*SDHB* /NC for 48 h) for 72 h. N = 3/group. **x, y** Quantification (**x**) and representative images (**y**) of α-SMA staining in the LX-2 cells with the same treatment as in (**w**). N = 3/group. Data are presented as mean ± SE. Significant differences are analyzed using two-tailed *t*-test (**t, u, w, x**), and a one-way ANOVA with Benjamini multiple comparisions (**b, d–g, i–q**), *n.s.* no significance. N values in (**b–r**) indicate the numbers of mouse and N values in (**t–y**) indicate biological independent replicates. Source data are provided as a Source Data file.

---

to the further deterioration of complex II activity caused by NEK7 deficiency.

Taken together, we found that NEK7 deficiency in hepatocytes caused HSC activation and exacerbated liver fibrosis induced by CCl₄, which could be reversed by inhibiting RET or ROS scavenging.

## NEK7 overexpression alleviates liver fibrosis
We first investigated the level of NEK7 in liver tissues and mitochondria from fibrotic mice and cell models. The expression of NEK7 was significantly reduced in the total and mitochondrial lysates from the mouse fibrotic livers induced by CCl₄ (Supplementary Fig. 10a–f), as

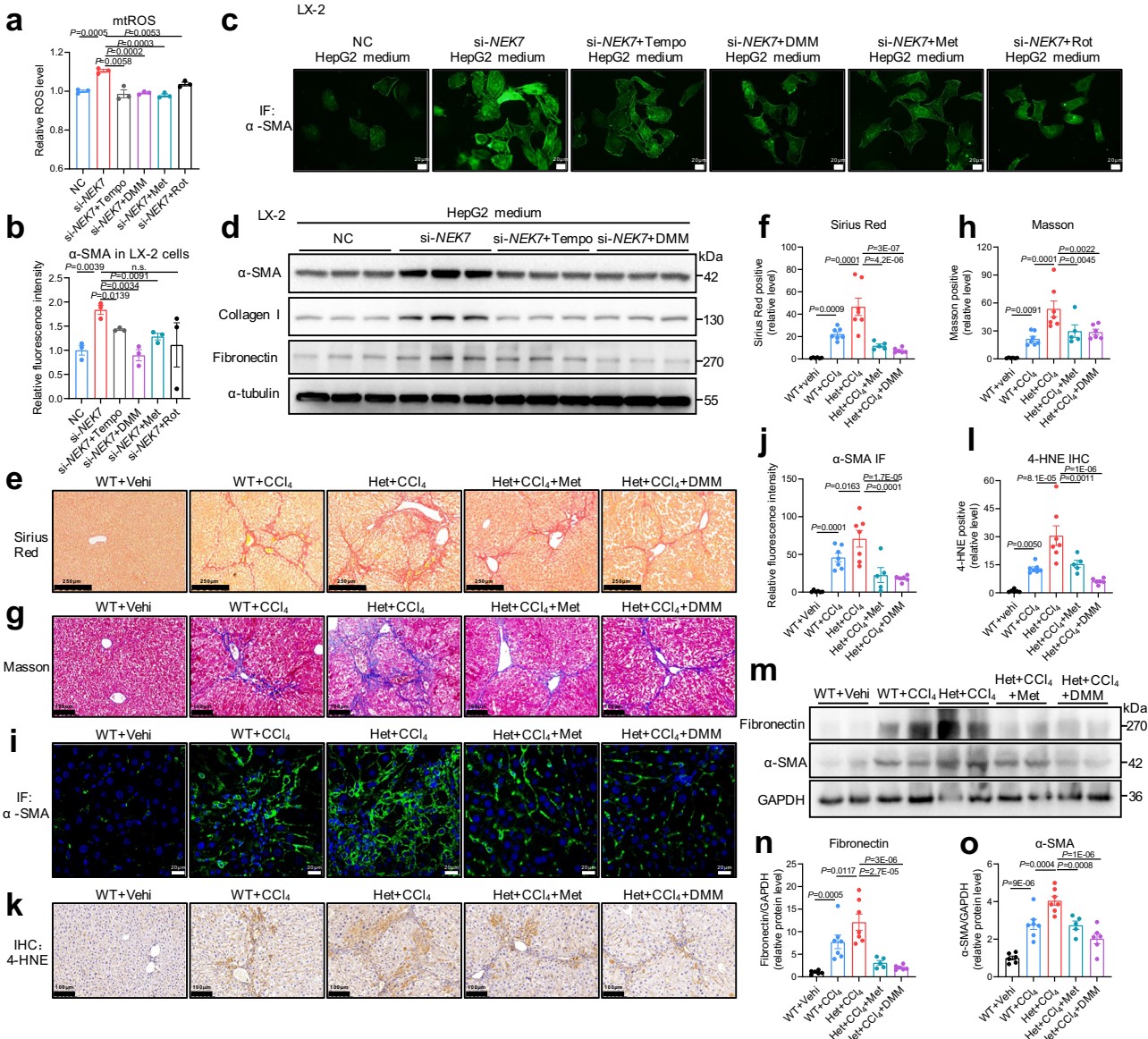

**Fig. 5 | Inhibiting RET alleviates the aggravation of CCl₄-induced liver fibrosis caused by NEK7 knockdown. a** Flow cytometry detected mitochondrial ROS levels (MitoSox Red) in HepG2 cells. The HepG2 cells are transfected with NC or si-*NEK7* for 48 h, then treated with 500 µM Tempo, 10 µM Dimethyl malonate (DMM), 100 µM Metformin (Met), or 50 nM Rotenone (Rot) for 24 h, respectively. N = 3/ group. **b, c** Relative fluorescence intensity (**b**) of α-SMA staining in LX-2 cells, the LX-2 cells are treated with the corresponding conditional medium of the HepG2 cells in (**a**). Representative images are displayed in (**c**). Scale bar, 20 µm. N = 3/ group. **d** Western blotting analysis of the expression of α-SMA, Collagen l, and Fibronectin protein in the LX-2 cells treated with the conditional medium of HepG2 cells. The HepG2 cells are transfected with NC or si-*NEK7* for 48 h, then treated with 500 µM Tempo and 10 µM DMM for 24 h, respectively. N = 3/group. The quantification in Supplementary Fig.8a–c. **e–l**, Representative images and quantification of

Sirius Red (scale bar, 200 µm), Masson (scale bar, 100 µm), α-SMA IF (scale bar, 20 µm), and 4-HNE immunohistochemistry (IHC) staining (scale bar, 100 µm). **m–o** Western blotting analysis showing the levels of fibrotic proteins (Fibronectin and α-SMA) in the livers of the mice with diverse treatments (see "Methods"). Quantification of the protein levels of Fibronectin and α-SMA displayed in (**n**, **o**). Data are presented as mean ± SE. Significant differences are analyzed using unpaired two-tailed *t*-test (**a**, **b**), and a one-way ANOVA with Benjamini multiple comparisions (**f**, **h**, **j**, **l**, **n**, **o**; WT+Vehi, *n* = 6; WT+CCl₄, *n* = 7; Het+CCl₄, *n* = 7; Het +CCl₄+Met, *n* = 5; Het+CCl₄ + DMM, *n* = 6). N values in (**a–d**) indicate biological independent replicates and N values in (**e–o**) indicate the numbers of mouse. Related data shown in Supplementary Fig. 8 and Supplementary Fig. 9. Source data are provided as a Source Data file.

well as in the total and mitochondrial lysates of primary hepatocytes extracted from the fibrotic livers (Supplementary Fig. 10a, g–j). In addition, NEK7 was also significantly downregulated in the mitochondria of the mouse fibrotic livers induced by CDAHFD (Supplementary Fig. 10k,l). Furthermore, the level of NEK7 in the fibrotic livers of NAFLD patients (fibrosis stage:4) also showed a decrement with a p-value of 0.0909 as analyzed by a two tailed t-test and a p-value of 0.0455 by a one tailed t-test (Supplementary Fig. 10m). In human-derived HepG2 cells treated with CCl₄ and free fatty acids (FFA,

palmitic acid/ oleate), the expression of NEK7 both in the total and mitochondrial lysates was also significantly downregulated (Supplementary Fig. 10n–u). Given the critical role of NEK7 in maintaining normal mitochondrial homeostasis in hepatocytes, the reduction of NEK7 could contribute to the pathogenesis and advancement of liver fibrosis.

To investigate the effect of NEK7 overexpression on chronic liver disease, we constructed a liver fibrosis model in the WT mice by intraperitoneal injection of CCl₄ for 6–7 consecutive weeks, during

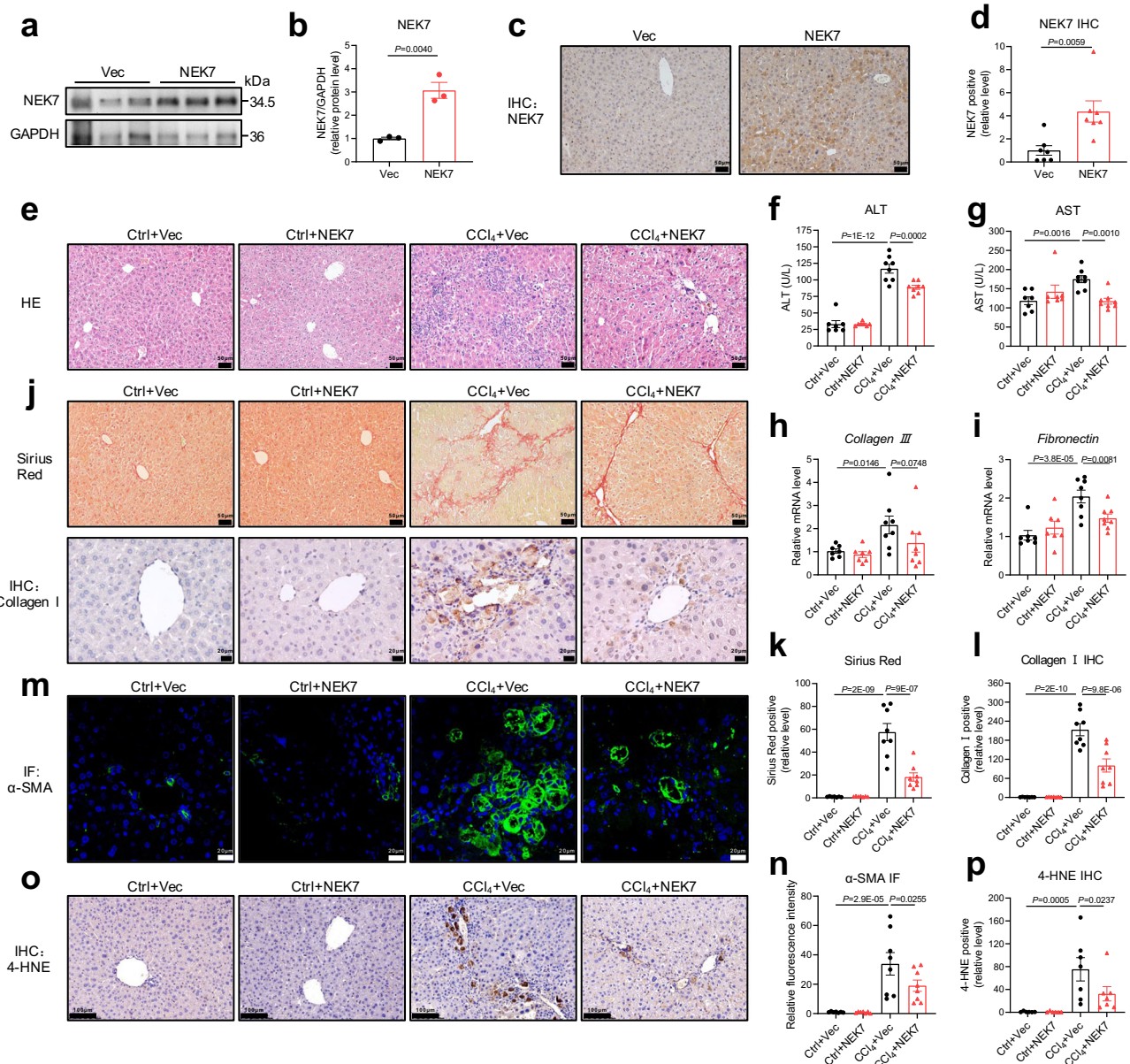

**Fig. 6 | NEK7 overexpression by tail vein injection of NEK7 plasmids alleviates CCl₄-induced liver fibrosis. a** Western blotting analysis showing NEK7 expression in the livers of the mice injected with vector (pcDNA3.1) or NEK7 (pcDNA3.1-mNEK7) plasmids. The quantification is in (**b**), $n = 3$/group. **c**, **d** Representative images and quantification of NEK7 IHC staining (scale bar, 50 μm), $n = 7$/group. **e** Representative images of HE staining (scale bar, 50 μm); Ctrl+Vec, $n = 7$; Ctrl +NEK7, $n = 7$; CCl₄+Vec, $n = 8$; CCl₄ + NEK7, $n = 8$. **f**, **g** Levels of serum ALT and AST in the mice with diverse treatments (see "Methods"); Ctrl+Vec, $n = 7$; Ctrl+NEK7, $n = 7$; CCl₄+Vec, $n = 8$; CCl₄ + NEK7, $n = 8$. **h**, **i** Relative mRNA levels of *Collagen* III and *Fibronectin* in the livers of the mice with diverse treatment (see "Methods"); Ctrl

+Vec, $n = 7$; Ctrl+NEK7, $n = 7$; CCl₄+Vec, $n = 8$; CCl₄ + NEK7, $n = 8$. **j–l** Representative images and quantification of Sirius Red (scale bar, 50 μm) and Collagen I IHC staining (scale bar, 20 μm); Ctrl+Vec, $n = 7$; Ctrl+NEK7, $n = 7$; CCl₄+Vec, $n = 8$; CCl₄ + NEK7, $n = 8$. **m–p** Representative images and quantification of α-SMA (scale bar, 20 μm; Ctrl+Vec, $n = 7$; Ctrl+NEK7, $n = 7$; CCl₄+Vec, $n = 8$; CCl₄ + NEK7, $n = 8$) and 4-HNE staining (scale bar, 100 μm; Ctrl+Vec, $n = 6$; Ctrl+NEK7, $n = 6$; CCl₄+Vec, $n = 7$; CCl₄ + NEK7, $n = 7$). Data are presented as mean ± SE. Significant differences are analyzed using unpaired two-tailed *t*-test (**b**,**d**), and a one-way ANOVA with Benjamini multiple comparisions (**f–i**, **k**, **l**, **n**, **p**). N values in (**a–p**) indicate the numbers of mouse. Source data are provided as a Source Data file.

which NEK7 plasmids (pcDNA3.1-mNEK7) were injected into the tail vein every six days. Injection of the NEK7 plasmids increased the expression of NEK7 in the liver tissue (Fig. 6a–d), which has been reported as an effective way of modulating the level of the indicated protein in the liver[43,44]. After a series of analysis, we found that decreased liver function, severe pathological injury, and fibrotic deposition induced by CCl₄ were all significantly alleviated in the NEK7 overexpressed mice (Fig. 6e–l). Additionally, CCl₄-induced proliferation and activation of the hepatic stellate cells and oxidative damage were also inhibited by NEK7 overexpression (Fig.6m–p).

To confirm that the protective role of NEK7 is hepatocyte-specific, we further used AAV8-TBG vector to deliver *NEK7* into the hepatocytes followed by CCl₄ intraperitoneal injection to induce liver fibrosis. The protein of NEK7 was overexpressed efficiently as evidenced by the western blotting and IF staining (Fig. 7a–d). Consistent with the protection of NEK7-plasmids tail vein injection, impaired liver function, fibrosis, activation of the hepatic stellate cells, inflammation and oxidative stress induced by CCl₄ were all markedly alleviated in the mice with NEK7-AAV (AAV8-TBG-NEK7-EGFP) treatment (Fig.7e–m,o–v). Additionally, we observed that the damaged mitochondria by CCl₄

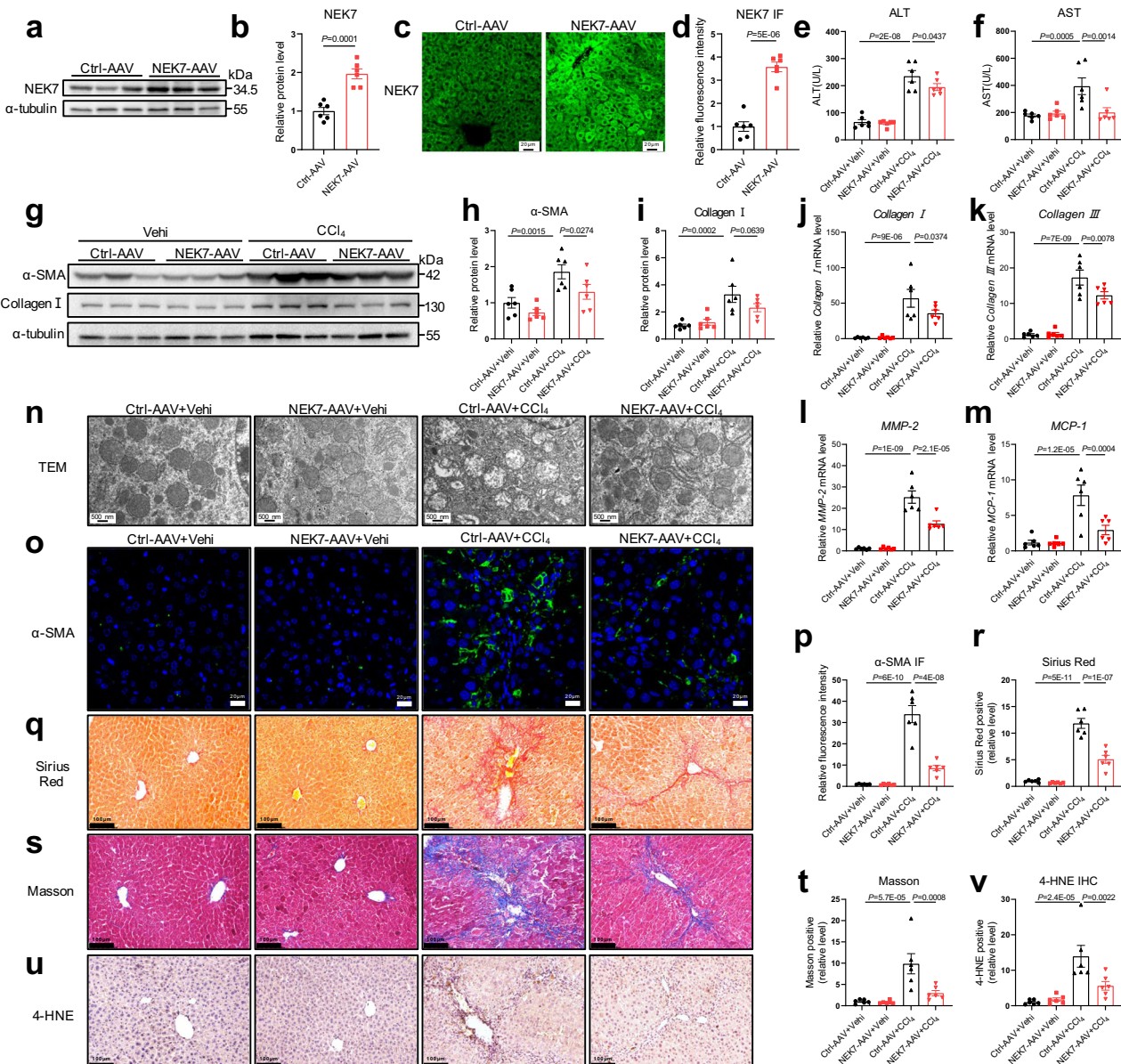

**Fig. 7 | NEK7 overexpression specific in hepatocytes by AAV8-TBG-NEK7 alleviates CCl₄-induced liver fibrosis. a, b** Western blotting analysis showing NEK7 expression in the livers of the mice injected with NEK7-AAV (AAV8-TBG-NEK7-EGFP) or Ctrl-AAV (empty vector). The quantification is in (**b**). **c, d** Representative images and quantification of NEK7 IF staining (scale bar, 20 μm). **e, f** Levels of serum ALT and AST in the mice with diverse treatments (see "Methods"). **g** Western blotting analysis of fibrotic protein levels (including Collagen I and α-SMA) in the livers of the mice. The quantifications are displayed in (**h, i**). **j−m** Relative mRNA levels of *Collagen I, Collagen* III*, MMP-2, and MCP-1* in the livers of the mice with diverse treatment (see "Methods"). **n** Representative TEM images depicting the ultrastructure of the mitochondria in hepatocytes from each group (scale bar, 500 nm), *n* = 4/group. **o−t** Representative images and quantification of α-SMA IF (scale bar, 20 μm), Sirius Red (scale bar, 100 μm), and Masson staining (scale bar, 100 μm). **u, v** Representative images and quantification of 4-HNE IHC staining (scale bar, 100 μm). Data are presented as mean ± SE. N value indicates independent replicates (from distinct samples). Significant differences are analyzed using unpaired two-tailed *t*-test (**b, d** *n* = 6/group), and a one-way ANOVA with Benjamini multiple comparisions (**e, f, h−m, p, r, t, v**, *n* = 6/group). N values in (**a−v**) indicate the numbers of mouse. Source data are provided as a Source Data file.

were also restored obviously, manifested in recovery of the mitochondrial morphology and cristae density, and the reduction of vacuolar degeneration (Fig. 7n).

Furthermore, we constructed a non-alcoholic fatty liver disease (NAFLD) model in the WT mice by feeding them choline-deficient, L-amino acid-defined, high-fat diet (CDAHFD) for four weeks. The NEK7 plasmids were also injected through the tail vein every six days to up-regulate the levels of NEK7 in the liver (Fig. 8a), consistent with the above model. We observed a similar protective effect of NEK7 on CDAHFD-induced liver fibrosis, as evidenced by decreased levels of fibrosis-related genes and proteins (Fig. 8b-h), decreased fiber deposition (Fig. 8i-l), and mitigated pathological injury and oxidative damage (Fig. 8m-o), although hepatic lipid accumulation (TG, triglyceride) was not affected (Fig. 8p). Taken together, NEK7 overexpression alleviated liver fibrosis induced by CCl₄ and CDAHFD, suggesting that NEK7 may be a promising therapeutic target for the treatment of chronic liver fibrosis.

## NEK7 is dispensable for the activation of NLRP3 inflammasome in hepatocytes

NEK7 is considered as an essential component for initiating the assembly and subsequent activation of NLRP3 inflammasome in

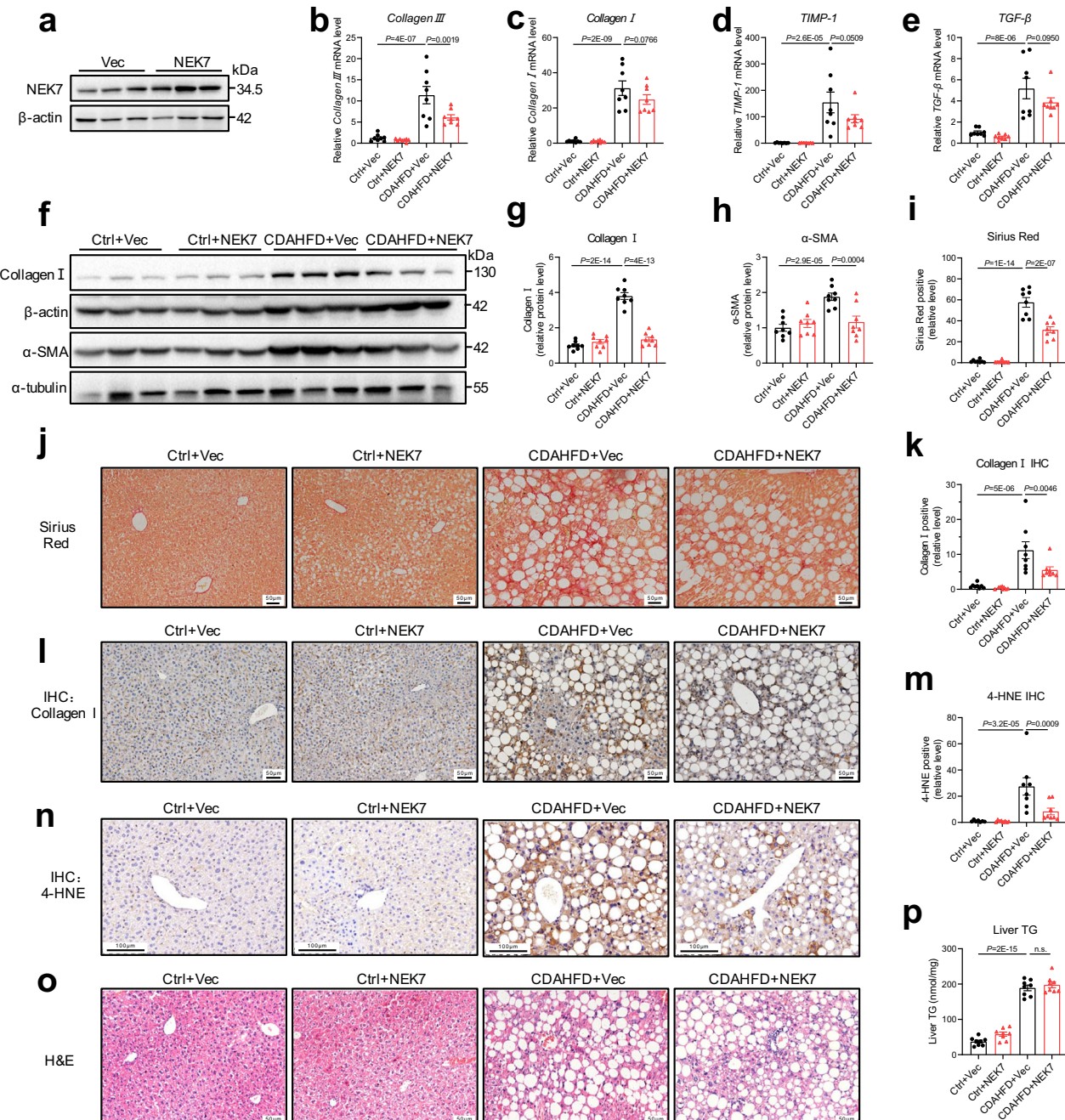

**Fig. 8 | NEK7 overexpression alleviates CDAHFD-induced liver fibrosis.**
**a** Western blotting analysis showing NEK7 expression in the livers of the mice injected with vector (pcDNA3.1) and NEK7 (pcDNA3.1-mNEK7) plasmids, $n = 3$/group.
**b**–**e** Relative mRNA levels of fibrotic genes (including *Collagen* III, *Collagen* I, *TIMP-1*, and *TGF-β*) in the mice with diverse treatments (see "Methods"); $n = 8$/group.
**f** Western blotting analysis of fibrotic protein levels (including Collagen I and α-SMA) in the livers of the mice. The quantifications are displayed in (**g**, **h**); $n = 8$/group.
**i**–**l** Representative images and quantification of Sirius Red (scale bar, 50 μm) and

Collagen I IHC staining (scale bar, 50 μm); $n = 8$/group. **m**, **n** Representative images and quantification of 4-HNE staining (scale bar, 100 μm); Ctrl+Vec, $n = 7$; Ctrl+NEK7, $n = 8$; CDAHFD+Vec, $n = 8$; CDAHFD + NEK7, $n = 8$. **o** Representative images of H&E staining (scale bar, 50 μm); Ctrl+Vec, $n = 7$; Ctrl+NEK7, $n = 8$; CDAHFD+Vec, $n = 8$; CDAHFD + NEK7, $n = 8$. **p** Triglyceride (TG) level in the liver tissues; $n = 8$/group. Data are presented as mean ± SE. Significant differences are analyzed using a one-way ANOVA with Benjamini multiple comparisions (**b**–**e**, **g**–**i**, **k**, **p**). N values in (**a**–**p**) indicate the numbers of mouse. Source data are provided as a Source Data file.

myeloid derived macrophages[16,17,45]. Our previous work suggested that the activation of NLRP3 inflammasome in APAP-induced liver injury was independent of NEK7[18]. In the present study, in CDAHFD-induced NAFLD mouse model, we found that the activation of NLRP3 inflammasome was not affected by overexpression of NEK7 in the liver via hydrodynamic injection of NEK7 plasmids, as evidenced by no significant difference in the expression of NLRP3 and caspase 1, the caspase 1 activity as well as the levels of mature IL-1β

in livers (Supplementary Fig. 11a–e). Furthermore, in the CCl₄-induced liver fibrosis model, overexpression of NEK7 specific in hepatocytes delivered by AAV8-TBG vector still did not affect the levels of NLRP3 inflammasome activation in the livers, consistent with the findings in the CDAHFD model (Supplementary Fig. 11f–j). These results indicate that, as in acute liver injury, the activation of NLRP3 inflammasome may also be independent of NEK7 in chronic liver injury.

Additionally, we further explored the level of NLRP3 inflammasome in *NEK7* knockout mice and their primary hepatocytes. In the *NEK7* knockout mice, although the expression of NLRP3 was slightly increased (Supplementary Fig. 11k–m), there was no significant increase or decrease in caspase 1 activity and IL-1β levels in the liver tissues (Supplementary Fig. 11n, o), similar to the results from the primary hepatocytes isolated from *NEK7* knockout mice (Supplementary Fig. 11p–s). Furthermore, to investigate whether NEK7 is indispensable for NLRP3 activation in hepatocytes, we treated the primary hepatocytes from the Het and WT mice with LPS and ATP (the classic stimuli for NLRP3 inflammasome activation) and found that the activation of NLRP3 inflammasome induced by LPS and ATP was not altered, as evidenced by no significant differences in the caspase 1 activity and mature IL-1β levels between groups (Supplementary Fig. 11t, u). Therefore, all these results confirm that NEK7 is dispensable for the NLRP3 inflammasome activation in hepatocytes in the present study.

## Discussion

Changes in the spatial conformation of the mitochondrial complex not only affect its activity but also the route or orientation of electron transfer, and finally determines the homeostasis of mitochondria and energy generation, which are involved in the development of various diseases[5,26,34]. There are still unknown molecules and mechanisms regulating the mitochondria that need to be explored. In the present study (Fig. 9), we revealed the localization and function of NEK7 in mitochondria for the first time; NEK7 enters the mitochondria, binds to SDHB, and promotes the spatial conformational stability of complex II, which is essential for maintaining the homeostasis of electron transport and energy generation in the mitochondria[25,26].

Mitochondrial targeting signal peptide is an important tool for targeting proteins to the mitochondria[20]. Generally, these peptides are located at the N terminal, with some exceptions reported in recent years, which are probably located at the C terminal or in the middle of the proteins[46,47]. In this study, we observed the co-localization of NEK7 and various mitochondrial markers through confocal imaging at high resolution, and confirmed the presence of NEK7 protein in the mitochondria isolated from different hepatocyte cell lines. This prompted us to analyze the protein sequence of NEK7. We observed two probable MTS inside it. Subsequent work has shown that both MTS could significantly target EGFP to the mitochondria, while exogenous NEK7 with MTS deletion could not enter the mitochondria. Therefore, these results strongly confirm the localization of NEK7 in the mitochondria. A recent study reported a new mechanism of NLRP3 inflammasome activation; the outer mitochondrial membrane protein March5 (also located in the endoplasmic reticulum adjacent to the mitochondria) in immune cells binds to the NACHT domain of NLRP3 and mediates its ubiquitination, promoting the recruitment of NEK7 and other members[48]. However, the evidence was too weak to indicate the localization of NLRP3 in the mitochondria, and no data showed the presence of NEK7 in the mitochondria.

An earlier study showed that *NEK7* whole-body knockout mice died in the embryonic stage or exhibited postnatal growth retardation, probably due to abnormal mitosis caused by *NEK7* deletion[49]. However, no other study has explored the phenotype of *NEK7* knockout mice. In this study, we bred *NEK7* hepatocyte specific knockout mice and found that all surviving Hom mice developed liver fibrosis at five months. Furthermore, in diverse hepatocyte cell lines and primary hepatocytes, we found that mitochondrial oxidative respiration was significantly inhibited and mitochondrial ROS production was abnormal after inhibiting NEK7, which may be an important cause of liver injury in *NEK7* hepatocyte knockout mice. We speculate that NEK7 may also play a role in the mitochondria of cells in other organs, and that NEK7 deletion-induced mitochondrial damage may be the key mechanism for the lethal phenotype of whole-body knockout mice, which needs to be further explored.

Mitochondrial ROS are generated by either forward electron transfer or RET, and the level of ROS from the latter is much higher than that from the former, which has been implicated in the development of diverse diseases[9,11–13,34]. Reverse electron transfer is mainly caused by an abnormal structure or substrate of mitochondrial complex II or I, which leads to an increase in coenzyme Q reduction and mitochondrial membrane potential[11,32]. Here, we found that the downstream molecule of NEK7 that regulates mitochondrial function is the SDHB subunit of mitochondrial complex II. However, NEK7 does not alter both the level of SDHB protein and complex II, while NEK7 directly binding to SDHB promotes the conformational stability of mitochondrial complex II, in which the electron transport distance increases slightly. When NEK7 is depleted, the stability of complex II deteriorates, and its activity is abnormally increased, which leads to over-reduction of coenzyme Q and an increase in membrane potential, thereby promoting the reverse transfer of electrons to complex I. Increased ROS generation in this process may explain why NEK7 deficiency in hepatocytes aggravates CCl₄-induced liver fibrosis. Additionally, abnormal complex II activity caused by CCl₄ could be restored by NEK7 overexpression, and inhibition of RET significantly alleviated liver fibrosis aggravated by NEK7 depletion, further verifying the above mechanism. Furthermore, we found that, in the HepG2 cells with NEK7 deficiency, the levels of succinate and L-glutamine, which are also important factors promoting the RET of complex II[35,37], were significantly increased. Although the specific mechanism is not clear, we speculate that NEK7 deficiency may also affect other metabolic pathways, or be secondary to mitochondrial dysfunction.

Fibrosis is a common pathological change that occurs during the progression of various chronic liver diseases, during which mitochondrial dysfunction and oxidative stress are important mechanisms. Therefore, targeting the mitochondria to treat chronic diseases is a promising therapeutic strategy in this field[50,51]. Mitochondrial RET has been shown to produce large amounts of ROS[9,12,13], which may be an important mechanism involved in fibrosis. In this study (Fig. 9), we found that the loss of NEK7 in hepatocytes leads to RET, which not only triggers spontaneous liver fibrosis but also aggravates the progression of CCl₄-induced hepatic fibrosis. Additionally, NEK7 was significantly reduced in hepatocyte and mitochondria from fibrotic livers. Furthermore, overexpression of exogenous NEK7 significantly improved chronic liver fibrosis and hepatic oxidative damage induced by both CCl₄ and CDAHFD, suggesting that the up-regulation of NEK7 in liver tissues may be a protective therapy, as the basal level of NEK7 under physiological conditions is insufficient for consolidating mitochondrial respiration function in diseased cells under hypoxia. However, this requires further exploration. In addition, we found that overexpression or inhibition of NEK7 did not affect the activation of NLRP3 inflammasome, as our previous findings in acute liver injury[18]. This further confirms that NEK7 is not indispensable for the activation of NLRP3 inflammasome under certain conditions, which have also been confirmed by more studies[52–54]. Therefore, this precisely prompts us to reveal the unexpected potential role of NEK7 in regulating mitochondrial function through complex II.

In conclusion, we found that NEK7 can enter the mitochondria via its MTS, to bind to SDHB, thereby promoting the spatial stability of complex II, and is essential for maintaining the homeostasis of mitochondrial electron transport and oxidative respiration. The important role of NEK7 in the mitochondria suggests that it may be a promising target in various mitochondrial oxidative stress-related diseases.

## Methods

### Mouse strains and model establishment

NEK7fl/fl (B6/JNju-Nek7em1Cflox/Nju, T004197), C57BL/6 J, and BALB/c mice were purchased from the Model Animal Research Center of Nanjing university (also known as GemPharmatech, Nanjing, China). The Alb-Cre mice (B6.Cg-Speer6-ps1Tg(Alb-cre)21Mgn/J, 003574), originally

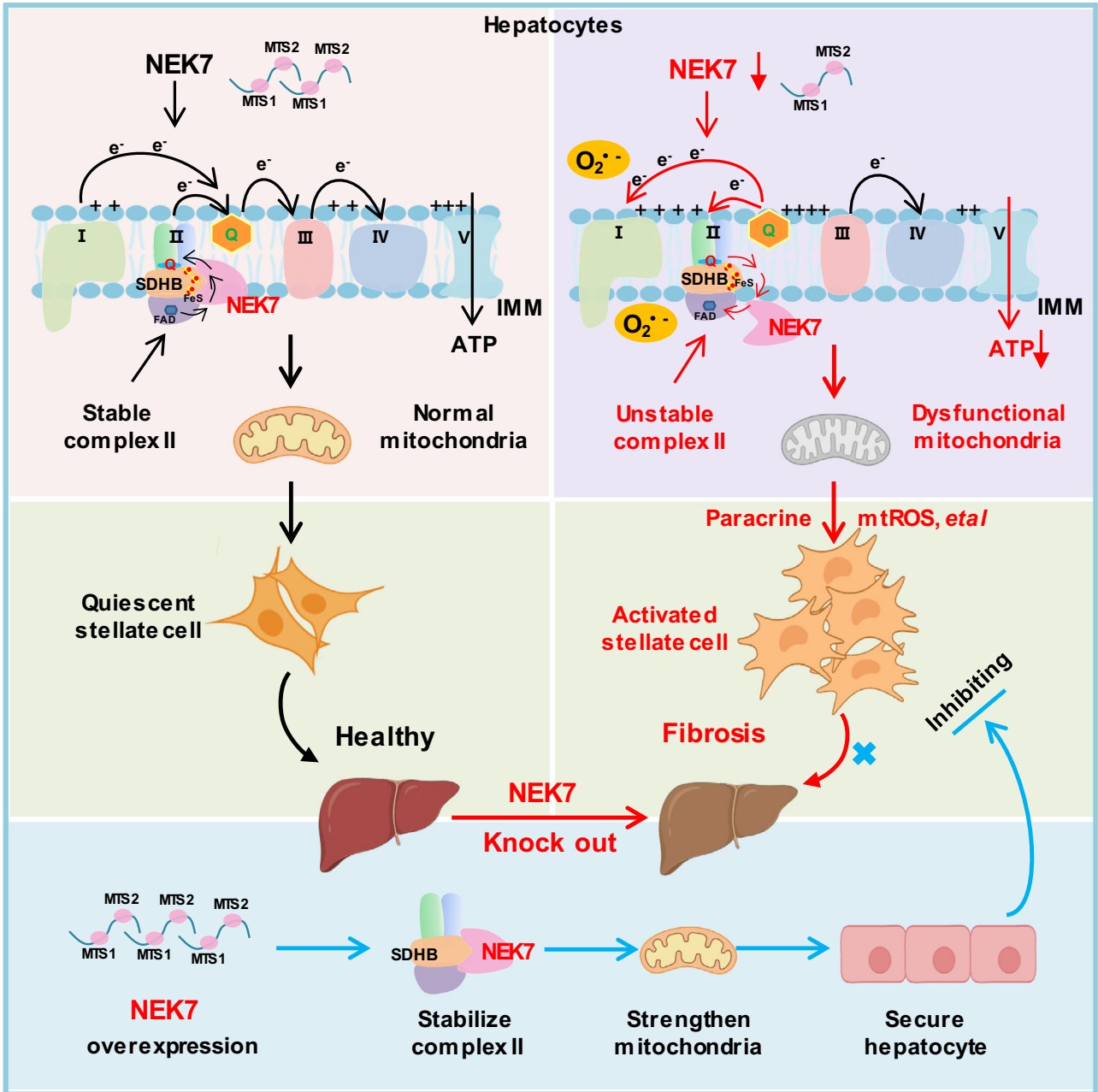

**Fig. 9 | Schematic of the roles of NEK7 in orchestrating the homeostasis of respiratory chain electron transport and impeding liver fibrosis progression.** Under normal conditions in hepatocytes, NEK7 protein, which is imported into mitochondria through its MTS, binds to SDHB, stabilizes the spatial conformation of mitochondrial complex II, and thus maintains the forward electron transfer in the mitochondrial respiratory chain. When NEK7 protein is reduced or absent in hepatocytes, SDHB cannot bind to NEK7, leading to changes in the spatial conformation of mitochondrial complex II, an increase in activity, and abnormal mitochondrial function, resulting in reverse electron transfer and large amounts of ROS generation, which then activate hepatic stellate cells through the paracrine effect, leading to liver fibrosis. Supplementation of exogenous NEK7 can stabilize mitochondrial complex II, strengthen mitochondrial function and significantly reduce liver fibrosis caused by various risk factors. *MTS* mitochondrial target signal; $e^-$ electron; $O_2^-$ superoxide anion; *ATP* Adenosine Triphosphate. Created in BioRender. Sun, z. (https://BioRender.com/l2fzi06).

derived from the Jackson Laboratory, were purchased from the Shanghai Model Organisms Center (Shanghai, China). Hepatocyte-specific *NEK7* knockout mice (*NEK7* cKO) were generated by crossing NEK7[fl/fl] mice with Alb-Cre mice. Reagents and primers used for genotyping were obtained according to the manufacturer's instructions.

All mice were maintained in a controlled specific pathogen-free environment with a 12 h light/12 h dark cycle (ambient temperature at 22–25 °C and humidity between 40 and 60%). Food and water were provided ad libitum. All the mice were euthanized at the end of the experiments under isoflurane anesthesia for subsequent analysis. All animal procedures were approved by the Institutional Animal Care and Use Committee (IACUC- 2209063) of Nanjing Medical University.

### Human liver tissues and sections
All the human liver tissues and sections used in this study were obtained from the normal liver tissues adjacent to the hepatoblastoma of patients (1#–8#) in the Children's Hospital of Nanjing Medical University, China. The clinical information including sex, age and diagnosis was included in Supplementary information Table 1. We performed co-localization staining of NEK7 and mitochondria (labeled

by COXIV) on the human liver tissue sections (patients 1#−5#), and detected the level of NEK7 in the extracted mitochondrial and cytoplasmic fractions from the human liver tissues (patients 6#−8#). The study protocol with the liver samples in this study was approved by the Committee on Clinical Research Ethics of Children's Hospital of Nanjing Medical University and informed consent was obtained from all participants (or their parents/guardians).

## Fibrosis models and treatment

To create liver fibrosis model, 8–12 weeks-old male BALB/c mice (randomly allocated) were either intraperitoneally (i.p.) injected with 15% $CCl_4$ ($CCl_4$/olive oil; from MACKLIN, China) twice a week for 6–7 weeks or fed with CDAHFD diet (45% fat with 1% cholesterol, choline-deficient; TP3622657, from Trophic Animal Feed High-Tech, China) for four weeks. The control mice were administered the corresponding vehicle in a similar manner. Blood and liver tissue samples were collected at the end of the experiment.

## Cell culture

The HepG2 cells (SUNNCELL, SNL-083, from ATCC) were cultured in Dulbecco's modified eagle medium (DMEM, Gibco) supplemented with 10% fetal bovine serum (PAN-biotech, Germany). HepaRG (Shanghai Hong Shun Biological Technology, C0986, from Millipore) and LX-2 (Procell, CL-0560) cells were maintained in RPMI 1640 medium (Gibco) supplemented with 10% fetal bovine serum (Gibco). Aml-12 cells (SUNNCELL, SNL-242, from ATCC) were cultured in DMEM-F12 medium (Gibco) supplemented with 10% fetal bovine serum (PAN-biotech). Subcultures were performed using 0.25% trypsin–EDTA (Gibco). All the cells were maintained in the cell incubator with constant temperature (37°C) and carbon dioxide (5%).

The primary hepatocytes used in this study were isolated from hepatocyte *NEK7* knockdown mice (the Het mice, male, 7-8 weeks) and the WT mice (male, 7–8 weeks) using a modified two-stage perfusion method. First, the mice were perfused through the inferior vena cava with 40 mL of P1 (Krebs Ringer buffer with 80 μL EGTA) for 6–7 min with the portal vein cut-off. Next, the mice were perfused with 40 mL of P2 (Krebs ringer buffer supplemented with 55 μL $CaCl_2$ and 8-10 mg collagenase IV, C8160, from Solarbio, China) for 6–7 min. Isolated livers were washed with PBS, digested, and dispersed in P2 buffer. The cell suspension was filtered with a 70 μM nylon mesh (biosharp, China), and re-suspended in percoll (Solarbio) for further purification by gradient centrifugation. The isolated hepatocytes were then stained with trypan blue to detect cell viability, and cultured in DMEM (Gibco) supplemented with 10% FBS. Detection was completed within 48 h of primary hepatocyte isolation.

## siRNA and plasmid transfection

siRNA transfection was performed with a mixture of 4−5 μL siRNA (20 μM, si-*NEK7* or si-*SDHB*, the target sequences in Supplementary Table 2) or negative control (RIBO-biotic, Shanghai) and 4−5 μL Lipofectamine 2000 reagent (11668019, Thermo Fisher Scientific), according to the manufacturer's methods. Plasmids (1−2 μg) were transfected into the cells by a ratio of 1:2 to Lipofectamine 2000. Plasmids used in this study, including pcDNA3.1-hNEK7, pcDNA3.1-mNEK7, pcDNA3.1-hSDHB, pcDNA3.1-hNEK7$^{\triangle MTS}$, pcDNA3.1-hNEK7$^{WT}$-3xFlag, pcDNA3.1-hNEK7$^{61-96}$-EGFP, pcDNA3.1-hNEK7$^{121-163}$-EGFP, pcDNA3.1-EGFP and pcDNA3.1, were synthetized in YouBio (Changsha, China). Protein levels were verified at 36-48 h after transfection.

## AAV8-TBG-cre-EGFP

AAV8-TBG-cre-EGFP (Adeno-Associated Viruses, Hanbio Biotechnology) was used to delete the hepatocyte NEK7. Specifically, 10-week-old male NEK7$^{fl/fl}$ mice (randomly allocated) were injected with 100 μL of AAV8-TBG-cre-EGFP ($1 \times 10^{12}$) or the control virus ($1 \times 10^{12}$) through tail vein. Four weeks later, the liver tissues were collected and embedded into OCT for frozen sectioning or prepared for western blotting.

## AAV8-TBG-NEK7-EGFP

AAV8-TBG-NEK7-EGFP (Adeno-Associated Viruses, Hanbio Biotechnology) was used to specifically overexpress NEK7 in hepatocytes. BALB/c mice (7–8 weeks old, male, randomly allocated) were injected with 100 μL of AAV8-TBG-NEK7-EGFP ($1 \times 10^{12}$) or the control virus ($1 \times 10^{12}$) through tail vein. Four weeks later, these mice were randomly grouped and then intraperitoneally (i.p.) injected with 15% $CCl_4$ ($CCl_4$/ olive oil; from MACKLIN, China) or the vehicle twice a week for 6–7 weeks to construct liver fibrosis model. Blood and liver tissue samples were collected at the end of the experiment.

## Hydrodynamic tail vein injection

Eight-week-old male BALB/c mice were used for hydrodynamic tail vein injections. The mice (randomly allocated) were administered 70 μg of pcDNA3.1-NEK7 plasmids (mouse, from Biogot, China) or empty plasmids (pcDNA3.1) every six days, which were diluted in 2 mL of saline and delivered as described in previous study[43,44]. The tail vein injection was performed as follows: Firstly, place the mouse on the tail vein injection device and expose its tail. Next, dilate the tail vein by wiping with alcohol, insert the syringe needle at an angle into the tail vein, and then inject the plasmids at a steady speed within 10 seconds.

## Flow cytometry

Mitochondrial ROS were detected using MitoSox Red (M36008, Thermo Fisher Scientific). The cells seeded in 12-well plates were incubated with MitoSox Red (1:1000 diluted in blank medium) for 15 min at 37 °C. MitoSox Red can penetrate the mitochondria and is specifically oxidized by superoxide. The cells were then washed with PBS and harvested for analysis using flow cytometry. CytExpert software (2.3.1.22) was used for post-acquisition analysis. For each sample, 10000 cells were recorded. FSC-A/SSC-A gating was used to determine the main cell population. FSC-A/FSC-H gating was used to exclude the doublets. Finally, the corresponding fluorescent signals (PE or FITC-Count) were analyzed.

The mitochondrial membrane potential was measured using TMRE assays (C2001S, Beyotime Biotechnology, Shanghai). At the indicated time point of the experiment, the cells were incubated with TMRE fluorescent probe (1:1000 diluted in blank medium) for 20 min at 37 °C, which can accumulate in the healthy mitochondria. The cells were then washed with PBS and harvested for analysis using flow cytometry. CytExpert software (2.3.1.22) was used for post-acquisition analysis. For each sample, 10000 cells were recorded. FSC-A/SSC-A gating was used to determine the main cell population. FSC-A/FSC-H gating was used to exclude the doublets. Finally, the corresponding fluorescent signals (PE or FITC-Count) were analyzed. Gating strategies are provided in Supplementary Fig. 12.

## Immunostaining

Liver tissues were immediately fixed with 4% paraformaldehyde after isolating from the mice, dehydrated, and embedded in paraffin. Paraffin-embedded slides were sequentially subjected to antigen retrieval and endogenous catalase blockade. After blocking, the slides were incubated with the primary antibody overnight, followed by incubation with the secondary antibody. Finally, the nuclei were stained with DAPI or hematoxylin. At the indicated time points, growing cells were fixed with paraformaldehyde and permeabilized with 0.2% Tritonx-100, followed by routine blocking, incubation with primary and secondary antibodies, and DAPI staining of the nucleus. Finally, the tissue sections or cells were observed under a fluorescence microscope (Olympus BX51) or laser confocal microscope (Zeiss 710).

For co-localization staining, we used MitoTracker Red CMXRos (MB6046, Meilunbio, Dalian), COXIV (66110-1-Ig, Proteintech) and TOM20 (A16896, ABclonal) antibodies as mitochondrial markers. The primary antibodies used here were from different species and were incubated simultaneously, according to the above immunostaining methods. High-resolution images were acquired using several laser confocal microscopes, including STELLARIS5 with LIGHTNING (Leica), csim110 (sunny, China), lsm900 with airyscan (Zeiss), and lsm710 (Zeiss).

The primary antibodies used here include rabbit anti-NEK7 (bs-7758R, BEIJING BIOSYNTHESIS), rabbit monoclonal anti-NEK7(ab133514, abcam), recombinant monoclonal mouse anti-SDHB(67600-1-Ig, Proteintech), rabbit monoclonal anti-a-SMA(ET1607-53, HUABIO), rabbit anti-Collagen I(bs-10423R, BEIJING BIOSYNTHESIS), rabbit anti-CD45 antibody (bs-4819R, BEIJING BIOSYNTHESIS), rabbit anti-EGFP(bs-2194R, BEIJING BIOSYNTHESIS) and 4-Hydroxynonenal Antibody(4-HNE, MAB3249, R&D). The secondary antibodies used were Alexa Fluor 647-labeled goat anti-mouse IgG (Beyotime, A0473) and Alexa Fluor 488-labeled goat anti-rabbit IgG (Beyotime, A0423). A universal two-step detection kit (PV-9000, mouse/rabbit enhanced polymer detection system; ZSGB-BIO) was used for immunohistochemical staining.

## Tissue fibrosis staining

To assess fiber deposition in the mouse liver, we performed Sirius red and Masson staining. Paraffin-embedded tissue sections were dewaxed and washed with ultrapure water. For the Sirius red staining, we stained the sections with Sirius Red dye (RS1220, G-CLONE, China) for 8–30 min. An identical staining time was used in each model. Masson staining was performed using a Masson staining kit (G1006, servicebio, China) referring to the manufacturer's instructions. Briefly, sections were sequentially stained with hematoxylin, Ponceau magenta, and 2.5% aniline blue dye. Images were acquired using an Olympus microscope or scanned with a digital pathology scanner (KFBIO).

## Co-immunoprecipitation (CO-IP)

The isolated mitochondrial proteins were used for co-immunoprecipitation assay, the concentration of which was at least 1 μg/μL. First, 200 μL protein lysis was incubated with NEK7 antibody (20 μg/μL, Abcam) or IgG antibody (30000-0-AP, Proteintech) for 2–4 h at room temperature. The antigen-antibody mixture was then incubated with agarose magnetic beads (B23202, Selleck) at 4 °C overnight. Next, the system was washed with a washing buffer and separated using a magnetic frame. Finally, protein loading buffer was added for denaturation at 95 °C, which were detected and analyzed by western blotting or mass spectrometry.

## Transmission electron microscopy (TEM)

Transmission electron microscopy was used to observe the ultra-structure of HepG2 cells and liver tissue. The cells ($1 \times 10^7$) or liver tissue (2–3 mm$^3$) were fixed with 2.5% glutaraldehyde, embedded, cut into ultrathin sections (50–70 nm), and counterstained with uranyl acetate and lead citrate. Finally, the sections were scanned under a transmission electron microscope (JEM-1400). RADIUS ALL 2.2 software was used for imaging and analysis.

## Subcellular fractionation

We isolated the subcellular fractions of HepG2 cells, AML-12 cells and human liver tissues to investigate the distribution of NEK7 proteins. At least $2 \times 10^7$ cells or 50 mg liver tissues were collected from each group and centrifuged. All the following steps were performed on ice, according to the instructions of the Mitochondria Isolation Kit (MCE, HY-K1060 and HY-K1061). A total of 2 mL isolation reagent (supplemented with PMSF) was added to the cell pellet or the tissues, homogenized 30 times using an ice bath homogenizer, and then centrifuged at $1000 \times g$ for 10 min. The precipitation was mainly the nucleus. After centrifuging the supernatant ($3500 \times g$ for 10 min), we obtained mitochondrial (precipitation) and cytosolic (supernatant) proteins. The protein concentration of each fraction was determined for western blotting.

## Blue native page (BN Page)

BN Page was performed to determine the assembly or integrity of complex II. Briefly, 60 μg of isolated mitochondria lysates were loaded on the 4–13% native page gels. Cathode buffer I (with 0.02% Coomassie Blue G-250, filled in the inner tank) and anode buffer (filled in the outer tank) was used for the electrophoresis at an initial voltage of 100 V. After 10–15 min, the cathode buffer I in the inner tank was replaced with the cathode buffer II (with 0.002% Coomassie Blue G-250). During the subsequent electrophoresis process, the voltage was gradually increased with the current below 50 mA until optimal separation. The gels were incubated in BN page transfer buffer (contain 0.1% SDS, BC600P, from Real-Times Biotechnology) for 5–10 min after electrophoresis, then the proteins in the gels were transferred to the PVDF membranes at 200 mA for 2 h. The entire process of electrophoresis and transfer were carried out in ice bath. Next, the membranes were incubated in Coomassie Blue dye for visualization and then washed, followed by blocking (5% defatted milk powder, 1 h), primary antibody incubation (anti-SDHB or anti-COX1, 4 °C, overnight), secondary antibody incubation (1 h), and image acquisition.

## Tempo, dimethyl malonate, metformin, and rotenone treatment

In vivo, we constructed a fibrosis model in hepatocyte *NEK7* knock-down mice (the Het mice) by CCl$_4$ injection (15%, i.p.). Two groups of mice were treated with dimethyl malonate (DMM, 200 mg /kg; HY-Y1787, MCE) or metformin (Met, 300 mg/kg; HY-17471A, MERCK) by gavage 2–3 h after every CCl$_4$-treatment. On the third day after the last injection of CCl$_4$, the mice were treated with DMM or Met. Three hours later, all the mice were anesthetized for sampling.

In vitro, we first inhibited the expression of NEK7 by si-*NEK7* transfection in the HepG2 cells. At 48 h, the HepG2 cells with NEK7 depletion were treated with 500 μM tempo (HY-W001187, MCE), 10 μM dimethyl malonate (DMM, HY-Y1787, MCE), 100 CM metformin hydrochloride (HY-17471A, MERCK), and 50 nM rotenone (HY-B1756, MCE), respectively. Twenty-four hours later, we transferred the medium from the HepG2 cells to the corresponding wells containing LX-2 cells. The LX-2 cells were cultured in conditioned medium for 72 h. Finally, both the HepG2 cells and LX-2 cells were collected for analysis.

## RNA-seq and mass spectrometry

For RNA-seq, 100 mg fresh liver tissue from each mouse (three WT and three NEK7 Hom mice) was snap-frozen in liquid nitrogen. All the samples were analyzed using DNBSEQ by BGI Genomics (China). The following KEGG analysis (Phyper in R software) and GO-C analysis (TermFinder package) were all performed on the Dr.TOM platform (BGI Genomics, China). We analyzed all the down-regulated genes ($\log_2$FC < 0, Q value < 0.05) in the livers of the Hom mice (compared to the WT mice) using KEGG pathway enrichment analysis, and then all the genes enriched in the top-ranked OXPHOS pathway were further undergone GO-C analysis.

For mass spectrometry, we first performed CO-IP with the NEK7 antibody (Abcam) and obtained CO-IP lysates in loading buffer, referred to the methods mentioned above. The following gel electrophoresis, proteolysis, HPLC and mass spectrometry detection were all performed by BGI Genomics (China). Briefly, after conventional gel electrophoresis, processed gel particles underwent proteolysis by trypsin (0.01 μg/μL) at 37 °C overnight. Next, peptide segments were obtained by gradient dissolution (dissolved in 50% and 100% ACN sequentially, centrifuge 5000 g for 1 minute, respectively). Finally, the resulting supernatant was centrifuged at 25,000 g for 5 minutes, and

the supernatant was taken for freeze-drying. Next, the dried peptide samples were reconstituted with mobile phase A (2% ACN, 0.1% FA), centrifuged at 20,000 g for 10 minutes, and then the supernatant was injected for peptide separation using Thermo UltiMate 3000 UHPLC with a self-packed C18 column, at a flow rate of 300 nL/min by the following effective gradient: 0 ~ 5 min, 5% mobile phase B (98% ACN, 0.1% FA); 5 ~ 45 min, mobile phase B linearly increased from 5% to 25%; 45 ~ 50 min, mobile phase B increased from 25% to 35%; 50 ~ 52 min, mobile phase B rose from 35% to 80%; 52 ~ 54 min, 80% mobile phase B; 54 ~ 60 min, 5% mobile phase B. Next, the separated peptides were detected by a tandem mass spectrometer Q-Exactive HF X (Thermo Fisher Scientific, San Jose, CA). The main parameters were set: ion source voltage was set to 1.9 kV; MS1 scanning range was 350 ~ 1500 m/z, resolution was set to 60,000; MS2 starting m/z was fixed at 100, resolution was 15,000. The iBAQ value of each protein was obtained using BGI's own software and the iBAQ algorithm. A total of 7980 peptides (1498 proteins) were identified. All the identified proteins were functionally annotated and analyzed using the Gene Ontology (GO) and Eukaryotic Orthologous Groups (KOGs) databases, and were involved in different functional categories, such as metabolism, cellular processes and signaling, information storage and processing, and other poorly characterized proteins. There are 68 proteins (refer to the Supplementary Data 1) pulldown by the NEK7 antibody that were enriched in the metabolism module, among which, three proteins (SDHB, TDIF2, and VATC1) were related to the OXPHOS pathway, whereas only SDHB is a known mitochondrial protein.

## Metabolite analysis

To detect the level of succinate, the HepG2 cells seeded in six-well plates were first transfected with si-*NEK7* (n = 3) or NC (n = 3) for 48 h. The cells were lysed and the supernatant is collected by extraction and centrifugation. Next, the cell lysates were immediately used for Elisa essay (YJ981747, Enzyme-linked Biotechnology, Shanghai), according to the manufacture's manual.

To detect the level of glutamine, the HepG2 cells seeded in 10-cm dish were first transfected with si-*NEK7* (n = 5) or NC (n = 5) for 48 h. The cells were harvested by trypsin digestion (2 min) and then centrifugated at 1500 g for 5 min. The following sample processing, LC-MS/MS analysis, chromatographic separation, and mass spectrometry data acquisition were all performed by BGI Genomics (China). Briefly, the precipitated cells were added 10 μL of internal standard and 800 μL of extraction reagent (methanol: acetonitrile: water, 2:2:1,) followed by grinding for 5 min. After precipitation for 2 h at −20 °C, the samples were centrifuged (25000 g,15 min) and followed by freeze-drying. Next, 120 μL of 50% methanol were added to dissolve the dried sample. After centrifugation (25000 g,15 min), 10 μL of the supernatant were used for the following LC-MS/MS analysis (Waters 2777c UPLC in series with Q exactive HF high-resolution mass spectrometer; USA). Chromatographic separation was performed on a Waters BEH C18 column (Waters, USA), and the column temperature was maintained at 45 °C. Primary and secondary mass spectrometry data acquisition were performed using Q Exactive HF (Thermo Fisher Scientific, USA).

## Seahorse

In this study, we performed seahorse assays on HepG2 cells, HepaRG cells, and primary hepatocytes (isolated from *NEK7* cKO mice). A total of 6000 cells were seeded in each well of the seahorse 96 well plates. The HepG2 and HepaRG cells were prepared for detection 48 h after the siRNA transfection. Primary hepatocytes were prepared for analysis within 48 h of isolation. The Mito stress test kit (Seahorse XF, Agilent Technologies) was used to evaluate mitochondrial respiration using a Seahorse XF96 Analyzer. Agents used for mitochondrial OCR detection include oligo (1 μM), FCCP (2–3 μM), antimycin A (0.5 μM),

and rotenone (0.5 μM). Data were acquired using an Agilent Seahorse Wave Pro 10.1.0 software.

## Mitochondrial complex activity

The activities of mitochondrial complex I, II, and V in the HepG2 cells and primary hepatocytes were determined according to the manufacturer's instructions (KTB1850/KTB1860/KTB1890; Abbkine). Briefly, a total of $1×10^7$ cells were collected and used for mitochondrial isolation. After precipitation and pretreatment, the mitochondria were added to the respective substrates for complex I, II, and V. Finally, the activity of each complex was reflected by the change in the substrates and metabolites, and detected using a microplate reader (Thermo Multiskan FC).

## Molecular docking and simulation of molecular dynamics

For molecular docking, both the structure of the NEK7 protein (PDB: 6S76) and SDH complex (PDB: 8GS8) were pretreated using Discovery Studio, and then virtual protein-protein docking was performed using ZDOCK module. Poses with ZDOCK scores of >15 were re-screened using RDOCK for precise docking analysis. PyMOL was used to visualize the three-dimensional binding sites.

For simulation of molecular dynamics (performed by Chengdu Mo-yan-suan Ltd., China), the simulation was conducted in multiple stages to ensure proper equilibration of the system. Initially, energy minimization was performed for 5000 steps using the steepest descent algorithm to eliminate any steric clashes between atoms. Subsequently, the system underwent a 100 ps NVT (isothermal-isochoric) equilibration at a constant temperature of 310.5 K. During this stage, the protein backbone atoms were restrained using a harmonic potential of 2.0 kcal/(mol·Å²) to facilitate the thorough relaxation of water molecules and ions, followed by a 1 ns NPT (isothermal-isobaric) equilibration at 310.5 K and 1 bar, during which the restraint force constant was progressively reduced to zero, enabling the system to attain a stable density. During both the equilibration and production phases, the temperature was maintained at 310.5 K using the V-rescale thermostat with a coupling constant of 0.2 ps. The pressure was regulated at 1 bar using the Parrinello-Rahman barostat with a coupling time of 0.2 ps. Long-range electrostatic interactions were computed using the Particle Mesh Ewald (PME) method, while a cutoff of 8.0 Å was applied for short-range non-bonded interactions. All covalent bonds involving hydrogen atoms were constrained using the SHAKE algorithm. The final production simulation was executed for 100 ns with a time step of 2 fs. The root mean square deviation (RMSD) and radius of gyration (rg) of the complex are analyzed for evaluating the equilibrium state, refer to Fig. 3m and Supplementary Data 3–5. The accuracy of the chosen models is sufficient to address the question under investigation, including all-atom model, explicit solvation model (SPC/E) and fixed-charge force field type (AMBER99SB-ILDN force field), also included in Table 1 in Supplementary Data 4. The major system setup, software used for simulation and analysis, and other parameters are included in Table 1 in Supplementary Data 4. The trajectory file of the molecular dynamic simulation has been provided as Supplementary Data 6. The distance of electron transport within the SDH complex (Fig. 3i) was measured after molecular dynamics simulations.

## Molecular interaction between NEK7 and SDHB

The human NEK7 (HY-P75935, MCE) and SDHB (Ag29868, Proteintech) proteins used in this study were commercially sourced recombinant proteins, and the purities of them were 94% and 80%, respectively. To elucidate the molecular interaction between NEK7 and SDHB, we performed Microscale thermophoresis (MST) on Monolith NT.115 (NanoTemper) and surface plasmon resonance (SPR) on Biacore T200 (GE Healthcare).

For MST analysis, one hundred-microgram of NEK7 (25 nM) and two hundred-microgram of SDHB (the highest concentration, 16.5 µM) protein were used. NEK7 proteins were labeled with the RED-NHS dye for capturing the variations in thermophoresis mobility. Finally, the dissociation constant (Kd) was obtained by a dose-response curve based on the concentration and normalized fluorescence intensity using MO.Affinity Analysis v2.3, offering a precise quantification of the proteins' interaction.

For SPR analysis, fifty-microgram of NEK7 or NEK7-mut and one hundred and eighty-microgram of SDHB protein were used. NEK7 or NEK7-mut proteins were flowed through the chip and were immobilized on it by amine coupling. Biacore T200 Control software was used to collect the data when different concentrations of the SDHB proteins were added to the reaction wells. The data was finally fit to a 1:1 Langmuir binding model to obtain the association and dissociation constants. The data from SPR analysis were provided in Supplementary Data 2.

### Icons or contents created in BioRender

The following icons or contents were created in BioRender. Sun, Z. (https://BioRender.com/l2fzi06); in Fig. 3a, mitochondrial membrane, proteins SDHA, SDHB, SDHC, SDHD, NEK7; in Fig. 4a, the mouse; in Fig. 4s, mitochondria, stellate cells; in Fig. 9, mitochondrial membrane, mitochondria, stellate cells, livers, hepatocytes. The use for publication has been granted by BioRender with a Publication License.

### Quantification and statistical analysis

All the data were presented as mean ± SEM and were analyzed using GraphPad Prism 10.0 software. Unpaired two-tailed Student's $t$-test was used to determine the statistical significance between two groups, and one-way ANOVA with Benjamini multiple comparison test was conducted to determine the statistical significance between every two groups in the experiments with more than three groups. Statistical significance was set at $p < 0.05$.

### Reporting summary

Further information on research design is available in the Nature Portfolio Reporting Summary linked to this article.

## Data availability

The raw RNA sequencing data generated in this study have been deposited at the NCBI GEO dataset under accession number GSE272141. The mass spectrometry data generated in this study have been deposited at ProteomeXchange Repository under accession number PXD068758. The full list of NEK7 interacting proteins (including the total 68 of NEK7-IP related proteins in energy production and conversion of the metabolism module) from the mass spectrometry analysis were provided in Supplementary Data 1. All other data are included in the Supplementary files and Source data are provided with this paper.

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

## Acknowledgements

We thank ELSEVIER for its language editing service during the preparation of this manuscript. We thank our colleagues for providing the procurement of reagents and consumables and the experimental guidance. We are thankful for the kind guidance from Yan Zhang on the isolation of primary hepatocytes. This work was supported by Social Development Fund of Jiangsu Province (BE2021607), "333"talent plan of Jiangsu province (333-2022001), and the China Postdoctoral Science Foundation (2022M721684). Illustrations in Figs. 3a, 4a, s, and 9 were created in PowerPoint (office 2019) with icons or contents created in BioRender. Sun, Z. (https://BioRender.com/l2fzi06).

## Author contributions

Z.J. and Z.S. conceptualized the project, designed the experiments. Z.J., A.Z., Y.Z. and Y.S. supervised this project. Z.S., L.S., H.H., Y.R. and W.Z. performed the experiments and analysis. Z.S., L.S., X.W., W.G., S.H. and D.Z. contributed to the investigation. Z.S. wrote the original draft. Z.J., A.Z., Y.Z. and Z.S. contributed to the writing– review & editing. The funding was acquired by A.Z. and Z.S.

## Competing interests

The authors declare no competing interests.
