## [Peer Review file · Nature Communications]

NEK7 couples SDHB to orchestrate respiratory chain electron transport homeostasis that impedes liver fibrosis

Corresponding Author: Professor Zhanjun Jia

Version 0:

Reviewer comments:

Reviewer #1

(Remarks to the Author)

In this study, Sun et al investigate the role of NIMA-related kinase 7 (NEK7) in mitochondria and demonstrate that NEK7 binds to succinate dehydrogenase complex iron sulfur subunit B, stabilizing the spatial conformation of this mitochondrial complex and preventing reverse electron transport leading to ROS generation. The authors were more ambitious to simply describing these mechanistic aspects of mitochondrial homeostasis provided by NEK7 and carried out experiments in animal models of liver fibrosis that could be relevant for potential fibrosis treatment. The study is of interest, but authors need to provide more convincing data, especially in the experimental models of liver fibrosis.

1. The introduction of the manuscript is quite comprehensive and describes previous data from the NEK7 hepatocyte-specific KO mice. However, it is not clear whether the RNA-seq data come from the previous publication in JHEP Reports or was de novo generated in this manuscript. Needs clarification.
2. As described in the introduction, NEK7 can directly bind to NLRP3 and initiate inflammasome assembly. However, this capacity was not explored in this study, which is directly linked to liver inflammation and fibrosis.
3. More details are needed in the bioinformatic analysis of the RNA-seq data, especially the results shown in Figure 1a. Did the authors replicated the data using GSEA? Also, please check the correctness of the units appearing in this figure.
4. Surprisingly, in Figure 1, once NEK7 has been linked to downregulation of oxidative phosphorylation in NEK7 hepatocyte-specific KO mice, the authors jump into the human describing the protein expression and the localization of NEK7 in a human hepatocyte cell line derived from a patient with hepatoblastoma. For consistency, it is necessary to perform these analyses in murine hepatocytes. This is also true for Figure 2, in which the role of NEK7 in the mitochondria is mainly explored in the human cell line and then confirmed in primary mouse hepatocytes, thus following a nonlogical pattern of data presentation. This is also true for Figure 3.
5. The finding that NEK7 hepatocyte-specific KO mice had a liver phenotype characterized by accelerated liver injury and fibrosis is key to translate this work to the human disease. However, the models of liver fibrosis rise some concerns. For example, injection of CCl₄ for 6-7 weeks typically renders mice with very advanced liver fibrosis almost approaching cirrhosis, which is a more advanced stage in the course of liver disease, with different clinical manifestations. In addition, the use of choline-deficient diet for 4 weeks appears to be too short and likely the mice only develop incipient fibrosis. Images in figure 7 show more liver steatosis than fibrosis.
6. To increase the impact of the paper, it would be nice to have some human data about the levels of NEK7 in patients with MASH.

Reviewer #2

(Remarks to the Author)

The manuscript by Sun et al. describes how NEK7 interacts with SDHB to regulate respiratory chain electron transport homeostasis, thereby mitigating liver fibrosis. This study, based on RNA sequencing of liver tissue from NEK7 hepatocyte-

knockout mice, reveals that NEK7 deficiency impairs mitochondrial oxidative phosphorylation and downregulates the expression of genes associated with the mitochondrial inner membrane. The mitochondrial localization of NEK7 may be mediated by mitochondrial targeting signal peptides, and its absence appears to induce mitochondrial dysfunction. Mice with liver-specific NEK7 knockout exhibit spontaneous liver dysfunction and fibrosis. While the findings may interest researchers in the field, the precise role of NEK7 in mitochondria remains unclear, and the preliminary nature of many of the data limits the ability to draw definitive conclusions. Specific questions are as follows:

Major:

- (1) NEK7 is known to play a key role in mitosis and the activation of the NLRP3 inflammasome (doi: 10.1038/ni.3333, doi: 10.1038/s41586-019-1295-z). However, in Fig. 1a, KEGG analysis of downregulated genes in NEK7 knockout mice did not reveal alterations in cell cycle or inflammation-related pathways. In contrast, Fig. 4 demonstrates a significant enhancement of inflammatory signaling in liver cells from NEK7 knockout mice. The authors are requested to clarify the rationale behind selecting downregulated genes for KEGG analysis and to explain why the analysis did not identify changes in cell cycle or inflammation-related pathways.
- (2) In Figs. 1c and 1d, GAPDH is detected in both the mitochondrial and cytoplasmic fractions. As GAPDH is present in both compartments, its use as an internal reference protein for the mitochondrial fraction does not provide a reliable indication of the separation efficiency between the two fractions. It is recommended to use ACTB, a cytoskeletal protein that is absent from the mitochondrial fraction, as an internal reference protein for the cytoplasmic fraction (doi: 10.1080/15548627). This would better assess the purity of the mitochondrial and cytoplasmic preparations.
- (3) In Fig. 1f, the expression level of NEK7- Δ MTS is notably lower than that of NEK7-WT, raising the possibility that the observed decrease in NEK7 mitochondrial co-localization may be due to reduced protein expression. Additionally, a significant difference in MitoTracker staining between the two groups could impact the accuracy of the co-localization quantification.
- (4) Fig. 3l,m: The binding of NEK7 and SDHB proteins has not been validated through reverse experimentation. It is recommended to introduce mutations at the predicted binding site and subsequently assess the direct interaction between the mutated proteins.
- (5) In Figs. 4o-q, liver-specific knockout of NEK7 triggers the activation of inflammation. However, other studies have reported that NEK7 promotes NLRP3-dependent cellular inflammatory responses (doi: 10.1038/ni.3333), and that NEK7 knockout alleviates chronic colitis in vivo (doi: 10.1038/s41419-019-2157-1). The authors are encouraged to include measurements of NLRP3 expression levels in the liver of NEK7 liver-specific knockout mice and to provide an explanation for the seemingly bidirectional effects of NEK7 knockout on the inflammatory response.
- (6) NOD-like receptor protein 3 (NLRP3) inflammasome activation occurs in Non-alcoholic fatty liver disease (NAFLD). Previous studies have demonstrated that inhibiting NLRP3 inflammasome activation reduces inflammatory recruitment and liver fibrosis in mouse models of steatohepatitis (doi: 10.1016/j.jhep.2017.01.022). However, NEK7 promotes NLRP3 assembly and activation, and the authors' overexpression of NEK7 in NAFLD mice appeared to alleviate liver fibrosis. The authors are encouraged to supplement their study with an assessment of inflammasome levels following NEK7 overexpression in NAFLD mice and to provide an explanation for the dual effects of NEK7 in the NAFLD model.

Minor:

- (1) Line130: The expression 'inhibition of NEK7 by siRNAs' is inappropriate.
- (2) Line152: The abbreviation TMRE should appear at Line139, Fig.2h.
- (3) Line148 has already shown heterozygous (Het), so Line233 does not need further explanation.

Reviewer #3

(Remarks to the Author)

I have reviewed the manuscript titled "NEK7 couples SDHB to orchestrate respiratory chain electron transport homeostasis that impedes liver fibrosis." The study explores a novel role for NEK7 in mitochondrial function by demonstrating its interaction with SDHB and its potential regulatory role in Complex II activity. The authors suggest that NEK7 deletion leads to mitochondrial dysfunction, increased ROS, and enhanced liver fibrosis, while NEK7 overexpression mitigates these effects.

This study presents an interesting perspective on the role of NEK7 in mitochondrial metabolism and liver fibrosis. However, there are several key concerns regarding the mechanistic depth of the findings, data interpretation, and experimental validation. While the proposed model is intriguing, additional evidence is needed to support the claim that NEK7 functions as a regulator of Complex II activity rather than as a structural scaffold. Furthermore, the specificity of DMM's effect in rescuing NEK7 deficiency has not been fully validated. Given these points, I believe that further experimental refinement is necessary before this work can be considered for publication in Nature Communications. Below, I outline major areas where additional data and clarifications would strengthen the manuscript.

Major Concerns

1. Insufficient Evidence Linking NEK7-SDHB Interaction to Complex II Functional Regulation

The central claim of the manuscript is that NEK7 directly modulates Complex II function through its interaction with SDHB. However, the authors do not provide sufficient experimental evidence to support this conclusion. While Co-IP and Mass Spectrometry data suggest an interaction between NEK7 and SDHB, this does not necessarily indicate a functional role.

- The study lacks direct measurements of Complex II enzymatic activity (e.g., SDH activity assay, succinate oxidation rate) in NEK7-deficient and overexpressing cells.

- It remains unclear whether NEK7 regulates Complex II enzymatic function or merely stabilizes SDHB as a scaffold protein.
- Post-translational modifications (PTMs) of SDHB (e.g., phosphorylation or ubiquitination) that could be influenced by NEK7 were not investigated.
- Blue Native PAGE or protein stability assays should be performed to confirm whether NEK7 affects Complex II assembly and integrity.

Conclusion: Additional functional assays would help establish whether NEK7 directly regulates Complex II enzymatic activity or if its role is primarily structural.

2. Lack of Evidence Supporting the NEK7-Specific Effect of Dimethyl Malonate (DMM) (Figure 5)

The authors propose that Dimethyl Malonate (DMM) rescues mitochondrial dysfunction and fibrosis in NEK7-deficient cells. However, it is unclear whether this effect is NEK7-specific or a general metabolic phenomenon related to SDH inhibition.

- DMM is a well-known SDH inhibitor and could exert protective effects through general metabolic modulation rather than NEK7-specific pathways.
- Control experiments using wild-type cells treated with DMM are necessary to determine whether DMM's effect is specific to NEK7 deletion.
- The authors do not show whether NEK7 regulates SDH activity directly. If DMM rescues fibrosis independently of NEK7 status, the authors' conclusions about NEK7's role are significantly weakened.

Conclusion: Further experiments are needed to confirm whether DMM's effects are directly linked to NEK7's function rather than being a general metabolic effect.

3. NEK7 Overexpression and its Functional Connection to Complex II Activity

- The authors provide phenotypic evidence that NEK7 overexpression reduces fibrosis, but they do not establish whether this is directly linked to Complex II function.
- If NEK7 regulates Complex II, its overexpression should enhance Complex II activity (measurable through SDH assays), but such data are missing.
- NEK7's potential role in mitochondrial structure or stability was not explored, leaving open the possibility that its effect on fibrosis is independent of Complex II activity.

Conclusion: Additional biochemical and structural analyses would help clarify the connection between NEK7 overexpression and Complex II function.

4. Lack of Clinical Relevance and Validation in Human Samples

To support the significance of their findings, the authors should consider validating their results in liver fibrosis patient samples.

- Is NEK7 downregulated in human liver fibrosis patients?
- Does NEK7 expression correlate with mitochondrial dysfunction markers in human liver samples?
- If NEK7-SDHB signaling is a conserved fibrosis-regulating pathway, it should be detectable in human samples.

Conclusion: Incorporating human sample validation would enhance the translational relevance of the study.

5. Lack of Liver-Specificity in pcDNA Injection and Potential Immune Effects

- The authors used pcDNA injection to overexpress NEK7, but this vector does not have liver specificity and may lead to NEK7 expression in immune cells as well.
- NEK7 is known to promote inflammasome activation in immune cells, which could result in increased immune activation rather than a direct antifibrotic effect in hepatocytes (DOI: 10.4049/jimmunol.208.Supp.52.02, 10.1126/sciimmunol.adl2993).
- To confirm that NEK7's effect is liver-specific, experiments should be conducted using a liver-specific promoter (e.g., Albumin promoter, AAV8/9-based delivery) to restrict expression to hepatocytes.

Conclusion: The use of pcDNA raises concerns that NEK7 overexpression may have influenced immune cells rather than acting specifically in hepatocytes. Liver-specific promoters should be used to verify that the observed effects are hepatocyte-driven.

Final Recommendation: Reject

Given the lack of direct evidence for NEK7-mediated Complex II regulation, the absence of functional validation in overexpression models, the potential confounding effects of DMM, the lack of liver specificity in pcDNA injection, and the lack of clinical validation, the current manuscript does not meet the high standards required for publication in Nature Communications.

I encourage the authors to refine their mechanistic insights and provide stronger experimental validation before resubmitting to a high-impact journal.

Reviewer #4

(Remarks to the Author)

Dear Authors, Thank you for a significant and important paper in the field of liver fibrosis. This reviewer find the work on NEK7 important.

This reviewer have a couple of improvements needed to recommend a publication - not enough detail is included:

Line 535-536: The description of the proteomics analysis is insufficient in order for others to reproduce the experiments. This must be corrected according to the MIAPE/HUPO guidelines. See the following link: (<https://www.psivdev.info/miape>),

<https://pubmed.ncbi.nlm.nih.gov/17687369/>.

Minimum requirements are: Information on sample preparation protocol, chromatographic separation (column brand, polarity, gradient, time, solvents), mass spectrometry settings and software used. It can be very short with referring to literature method, but the above information must be included.

Example of level of details of proteomics for liver disease can be viewed in papers like:
<https://pubmed.ncbi.nlm.nih.gov/35654907/>

Finally, at least a peak list of all of the 68 proteins pulldown by the CO-IP NEK7 must be in a supplementary file.

Line 540: From above – a peak list of the 68 protein must be included in supplementary file.

Line 551-552: Details on digestion, sample processing, metabolite extraction and UHPLC analysis (column, gradient) is missing. Needed in order for others to reproduce experiments.

Version 1:

Reviewer comments:

Reviewer #1

(Remarks to the Author)

The authors made a great job in preparing the revised version of the manuscript, which has significantly improved its quality. However, this reviewer is still not convinced of using cell lines instead of primary hepatocytes from mice, which are readily available after disaggregation of the liver tissue. This concern is related to comment number 4. This is my original comment: Surprisingly, in Figure 1, once NEK7 has been linked to downregulation of oxidative phosphorylation in NEK7 hepatocyte-specific KO mice, the authors jump into the human describing the protein expression and the localization of NEK7 in a human hepatocyte cell line derived from a patient with hepatoblastoma. For consistency, it is necessary to perform these analyses in murine hepatocytes. This is also true for Figure 2, in which the role of NEK7 in the mitochondria is mainly explored in the human cell line and then confirmed in primary mouse hepatocytes, thus following a nonlogical pattern of data presentation. This is also true for Figure 3.

It would be necessary to show data in primary murine hepatocytes instead of using AML-12 cells, unless the authors justify the technical reasons why they are not using the primary cell approach.

Reviewer #2

(Remarks to the Author)

The author has answered all my questions. I have no further questions for the author to respond to. I recommend that it be published in Nature Communications.

Reviewer #3

(Remarks to the Author)

Comments to the Authors

This manuscript investigates the role of NEK7 in mitochondrial electron transport, proposing that NEK7 stabilizes Complex II via binding to SDHB, thereby suppressing reverse electron transport (RET) and reducing ROS overproduction, which in turn alleviates liver fibrosis. The authors present both in vitro and in vivo data supporting that NEK7 overexpression can mitigate liver fibrosis induced by CCl₄ and CDAHFD, while NEK7 deficiency worsens it. This is an original and potentially impactful study, highlighting a new mitochondrial regulatory mechanism.

However, I have significant concerns about the translational and mechanistic validity of the study that, in my view, preclude publication at this stage.

1. The central hypothesis is that boosting NEK7 could be therapeutic for liver fibrosis. However, the authors do not show whether NEK7 is actually downregulated in human fibrotic liver samples, or in standard mouse models of liver fibrosis, prior to their interventions. Without this information, it is unclear whether NEK7 represents a physiologically relevant therapeutic target. If NEK7 is not reduced during fibrosis, enhancing its expression may have limited translational value.

2. Although the authors propose that NEK7 binding stabilizes Complex II conformation, it is paradoxical that NEK7 deficiency increases Complex II activity while destabilizing its structure. This mechanistic disconnect is only partially addressed, and remains speculative without more rigorous structural or in vivo functional data.

3. No human fibrotic tissue samples were analyzed for NEK7 levels. The only human data comes from normal tissue adjacent to hepatoblastoma, which does not reflect fibrosis pathophysiology. As a result, the clinical relevance of the findings is questionable.

Overall assessment

While the topic is interesting and the data collection appears technically sound, the above concerns represent major gaps. In particular, the lack of evidence that NEK7 is reduced in fibrosis models or in patients is a critical weakness that undermines the proposed therapeutic rationale. Therefore, I do not support publication of this manuscript in its current form.

Reviewer #4

(Remarks to the Author)

From this reviewers point of view: changes made in the revised manuscript were sufficient to recommend publication.

Version 2:

Reviewer comments:

Reviewer #1

(Remarks to the Author)

The authors have satisfactorily addressed all my previous concerns about the use of cell lines instead of primary liver cells.

Reviewer #3

(Remarks to the Author)

The revised manuscript has been improved significantly. The additional experiments strengthen the conclusions, and the authors have addressed the previous concerns. The study is now clear, convincing, and suitable for publication.

I recommend acceptance in Nature Communications.

Response to the Reviewers

Response to Reviewer #1:

In this study, Sun et al investigate the role of NIMA-related kinase 7 (NEK7) in mitochondria and demonstrate that NEK7 binds to succinate dehydrogenase complex iron sulfur subunit B, stabilizing the spatial conformation of this mitochondrial complex and preventing reverse electron transport leading to ROS generation. The authors were more ambitious to simply describing these mechanistic aspects of mitochondrial homeostasis provided by NEK7 and carried out experiments in animal models of liver fibrosis that could be relevant for potential fibrosis treatment. The study is of interest, but authors need to provide more convincing data, especially in the experimental models of liver fibrosis.

Response: We appreciate the thoughtful and constructive suggestions from the reviewers.

1. The introduction of the manuscript is quite comprehensive and describes previous data from the NEK7 hepatocyte-specific KO mice. However, it is not clear whether the RNA-seq data come from the previous publication in JHEP Reports or was de novo generated in this manuscript. Needs clarification.

Response: Thank the Reviewer for pointing out this confusion. Actually, the RNA-seq data were de novo generated in WT and Homozygous NEK7 KO mice in this study. In our previous study published in JHEP Reports, the RNA-seq data were from WT and heterozygous NKE7 KO mice. We have corrected these confused sentences in the '**Introduction**' and '**Results**' part.

2. As described in the introduction, NEK7 can directly bind to NLRP3 and initiate inflammasome assembly. However, this capacity was not explored in this study, which is directly linked to liver inflammation and fibrosis.

Response: Thanks for the Reviewer's valuable suggestions. As you mentioned, studies have shown that NLRP3 inflammasome are involved in liver inflammation and fibrosis¹⁻⁴. However, they only examined the protein level of NEK7 or the pull-down level of NEK7 immunoprecipitated by NLRP3, which are insufficient to demonstrate that NEK7 is indispensable for NLRP3 inflammasome activation under these conditions¹⁻⁴. Our previous work revealed that NEK7 is dispensable for NLRP3 inflammasome activation in APAP-induced acute liver injury⁵. Moreover, growing number of studies suggested other pathways involved in NLRP3 inflammasome activation independent of NEK7⁶⁻⁸.

In the present study, to better address this concern, we performed additional experiments

to explore the potential role of hepatocyte NEK7 in the activation of NLRP3 inflammasome in liver fibrosis, as shown in the following:

In CDAHFD-induced fibrotic livers, consistent with previous studies^{9,10}, the protein levels of caspase 1 and NLRP3 were increased and the NLRP3 inflammasome was significantly activated. However, the NLRP3 inflammasome activation was not influenced by NEK7 overexpression (**Supplementary Fig.10a-e**) in this study. Additionally, in the CCl₄-induced liver fibrosis model, the component level and the activity of NLRP3 inflammasome were also not influenced by NEK7 overexpression (**Supplementary Fig.10f-j**). Furthermore, we explored the component level and activity of NLRP3 inflammasome in the *NEK7* knockout mouse livers. Although the hepatic protein level of caspase 1 and NLRP3 increased slightly in the Het mice, the NLRP3 inflammasome were not activated both in the Het and the Hom mice livers (**Supplementary Fig.10k-o**), which were confirmed in the primary hepatocytes isolated from the *NEK7* knockout mouse livers (**Supplementary Fig.10p-s**). To further investigate whether NEK7 is indispensable in the activation of NLRP3 inflammasome, LPS (1 µg/ml) and ATP (2.5 mM), the classic stimulus combination that induces NLRP3 inflammasome activation¹¹⁻¹³, were used to treat the primary hepatocytes from the Het mouse livers. We found that combined treatment of LPS and ATP induced the NLRP3 inflammasome activation in the primary hepatocytes from the WT mice, which was not influenced by NEK7 knockdown (Het) (**Supplementary Fig.10t-u**).

Taken together, these data suggest that NEK7 is dispensable in the activation of NLRP3 inflammasome in hepatocytes in the present experimental settings.

Supplementary Fig.10

Supplementary Fig.10 Effects of NEK7 on the activation of NLRP3 inflammasome determined in diverse models.

a-c, Western blotting analysis showing the protein levels of NLRP3 and Caspase 1 in the mouse livers of the NEK7-CDAHFD model. The quantification is in **(b,c)**. N=8/group.

d, Relative level of Caspase 1 activity in the livers. N=8/group.

e, Relative level of IL-1 β in the livers tested by ELISA. N=8/group.

f-h, Western blotting analysis showing the protein levels of NLRP3 and Caspase 1 in the mouse livers of the NEK7-AAV-CCl₄ model. The quantification is in **(g,h)**. N=6/group.

i, Relative level of Caspase 1 activity in the livers. N=6/group.

j, Relative level of IL-1 β in the livers tested by ELISA. N=6/group.

k-m, Western blotting analysis showing the protein levels of NLRP3 and Caspase 1 in the livers from the NEK7 knockout mice. The quantification is in **(l,m)**. N=6/group.

n, Relative level of Caspase 1 activity in the livers. N=6/group.

o, Relative level of IL-1 β in the livers tested by ELISA. N=6/group.

p-r, Western blotting analysis showing the protein levels of NLRP3 and Caspase 1 in primary hepatocytes

isolated from the WT and Hom mice. The quantification is in (q,r). N=6/group.

s, Relative level of IL-1 β in the primary hepatocytes (from WT, Het, Hom) tested by ELISA. N=5/group.

t, Relative level of Caspase 1 activity in the primary hepatocytes. The primary hepatocytes isolated from the WT and Het mice livers were treated with LPS (1 μ g /ml) and ATP (2.5 mM) for 24 h. N=3/group.

u, Relative level of IL-1 β in the medium of the primary hepatocytes tested by ELISA. The hepatocytes were treated as shown in (t). N=3/group.

Data are presented as mean \pm SE. N value indicates independent replicates (from distinct samples). Significant differences are analyzed using unpaired two-tailed *t*-test (g-j,q,r), and a one-way ANOVA with Benjamini multiple comparisons (b-e,l-o,s-u).

3. More details are needed in the bioinformatic analysis of the RNA-seq data, especially the results shown in Figure 1a. Did the authors replicated the data using GSEA? Also, please check the correctness of the units appearing in this figure.

Response: Thank the Reviewer for pointing out this confusion. The analysis of the RNA-seq data in **Fig.1a**, including KEGG analysis (Phyper in R software) and GO-C analysis (TermFinder package) were all performed on the Dr.TOM platform (BGI Genomics, China). We analyzed all the down-regulated genes ($\log_2FC < 0$, Q value < 0.05) in the livers of the Hom group (compared to the WT mice) using KEGG pathway enrichment analysis, and then all the genes enriched in the top-ranked OXPHOS pathway were further undergone GO-C analysis. GSEA analysis was not used in this study.

We have revised the related descriptions in the '**Results**' and the '**Methods**' part, respectively. We also checked and displayed the full name of each unit in **Fig.1a**.

4. Surprisingly, in Figure 1, once NEK7 has been linked to downregulation of oxidative phosphorylation in NEK7 hepatocyte-specific KO mice, the authors jump into the human describing the protein expression and the localization of NEK7 in a human hepatocyte cell line derived from a patient with hepatoblastoma. For consistency, it is necessary to perform these analyses in murine hepatocytes. This is also true for Figure 2, in which the role of NEK7 in the mitochondria is mainly explored in the human cell line and then confirmed in primary mouse hepatocytes, thus following a nonlogical pattern of data presentation. This is also true for Figure 3.

Response: We agree with the Reviewer. Following the Reviewer's suggestions, we performed additional experiments in AML-12 cells and made an adjustment of the data presentation (shown in the following details and the corresponding changes in the revised manuscript and figures).

In **Fig.1** and **Supplementary Fig.1**, we performed co-localization staining of NEK7 and

MitoTracker in AML-12 cells (**Supplementary Fig.1a,b**), and made an adjustment of the data presentation in the 'Results'.

In **Fig.2**, we put all the results from the mouse liver and primary hepatocytes in front of the results from the human cells (HepG2 and HepaRG), and then adjusted the descriptions of the data presentation in the 'Results'.

In **Fig.3** and **Supplementary Fig.4**, we performed additional CO-IP experiments (**Supplementary Fig.4a**) and co-localization staining of NEK7 and SDHB (**Supplementary Fig.4b,c**) in AML-12 cells, and then adjusted the descriptions of the data presentation in the 'Results'.

Supplementary Fig.1 Co-localization analysis of NEK7 and mitochondria

a,b, Representative images acquired by laser confocal microscopy depicting NEK7 distribution (Green) in the mitochondria (Red, MitoTracker) of AML-12 cells. Scale bar, 5 μ m. Pearson's Rr and Overlap R (**b**) analyzed by the ImageJ plugin are 0.814 and 0.849, indicating a high co-localization of NEK7 with mitochondria. N=3.

Supplementary Fig.4 Validation of the interaction of NEK7 and SDHB in AML-12 cells.

a, Western blotting analysis showing the interaction of NEK7 and SDHB determined by co-immunoprecipitation with IgG and NEK7 antibody.

b,c, Representative images (**b**) of immunofluorescence show NEK7 and SDHB co-localization in AML-12 cells. Scale bar, 5 μ m. The ImageJ plugin conducts co-localization analysis (**c**), including Pearson's correlation Rr and Overlap R. N=3.

5. The finding that NEK7 hepatocyte-specific KO mice had a liver phenotype characterized by accelerated liver injury and fibrosis is key to translate this work to the human disease. However, the models of liver fibrosis raise some concerns. For example, injection of CCl₄ for 6-7 weeks typically renders mice with very advanced liver fibrosis almost approaching cirrhosis, which is a more advanced stage in the course of liver disease, with different clinical manifestations. In addition, the use of choline-deficient diet for 4 weeks appears to be too short and likely the mice only develop incipient fibrosis. Images in figure 7 show more liver steatosis than fibrosis.

Response: Thanks for the Reviewer's comments. In this study, we administered the mice with CCl₄ (15%) twice a week for 6-7 weeks, which is a classic regimen for a liver fibrosis model and has been widely used and validated in numerous studies¹⁴⁻¹⁶. Actually, to reach the cirrhosis stage, more frequent (three times a week), higher doses (25-50%), or a longer duration (12 weeks) of administration are required¹⁴⁻¹⁶.

To construct NAFLD model in this study, the mice were fed freely with CDAHFD (choline-deficient, amino acid-defined, 45% fat with 1% cholesterol, for 4 weeks) diet, which is a modified MCD diet. Compared to the traditional MCD diet, besides the defined amino acid, the CDAHFD contains a higher proportion of fat and cholesterol, which induce rapid liver fibrosis with more pronounced steatosis, better simulating human NAFLD^{10,17,18}. Previous study has shown that significant liver fibrosis can be induced within just three weeks in the mice fed with CDAHFD¹⁸. Therefore, in the present study, above models were employed and the fibrotic response was examined in the experimental settings.

6. To increase the impact of the paper, it would be nice to have some human data about the levels of NEK7 in patients with MASH.

Response: Thanks for the Reviewer's suggestion. We apologized for not being able to obtain adequate samples of MASH in the short term. As you know, we are from a children's hospital (Children's Hospital of Nanjing Medical University), in which almost no MASH children underwent the liver biopsy. However, we have liver samples from hepatoblastoma children and examined the subcellular localization of NEK7 in hepatocytes using the paraffin-

embedded sections derived from normal liver tissues adjacent to the hepatoblastoma in patients from our hospital, which showed a high localization of NEK7 in COXIV-labeled mitochondria (**Supplementary Fig.1f**). In addition, we extracted mitochondria and cytoplasm fractions from normal liver tissue adjacent to the tumor and detected the level of NEK7. We found that there is also a high abundance of NEK7 in the human hepatic mitochondria (**Supplementary Fig.1g**), consistent with the results from the hepatocyte cell lines and mouse liver tissues in the manuscript, further confirming the distribution of NEK7 in hepatocyte mitochondria. Due to the high conservation of NEK7 between human and mouse, combined with the important role of NEK7 in mitochondria of human hepatocyte lines investigated in this study, it strongly suggests that NEK7-orchestrated mitochondrial complex II is crucial for the maintenance of mitochondrial homeostasis and normal hepatocyte function. Undoubtedly, further exploration using appropriate samples of human chronic liver disease would bring more benefit for clinical translation, which is also the limitation of this study.

Supplementary Fig.1 Co-localization analysis of NEK7 and mitochondria.

f, Representative images of the co-localization staining of NEK7 (Green) and mitochondria (COXIV, Red) in human liver tissues. Scale bar, 20 μm. N=5 (patient 1#-5#, shown in **Supplementary information Table 1**).

g, Western blotting analysis showing the NEK7 protein levels in mitochondria and cytoplasm isolated from human liver tissues (N=3, patient 6#-8#, shown in **Supplementary information Table 1**). Mito, mitochondria; Cyto, cytoplasm.

References

- 1 de Carvalho Ribeiro, M. & Szabo, G. Role of the Inflammasome in Liver Disease. *Annu Rev Pathol* **17**, 345-365, doi:10.1146/annurev-pathmechdis-032521-102529 (2022).
- 2 Sheng, M. *et al.* CD47-Mediated Hedgehog/SMO/GLI1 Signaling Promotes Mesenchymal Stem Cell Immunomodulation in Mouse Liver Inflammation. *Hepatology* **74**, 1560-1577, doi:10.1002/hep.31831 (2021).
- 3 Liu, J. *et al.* CLICs Inhibitor IAA94 Alleviates Inflammation and Injury in Septic Liver by Preventing Pyroptosis in Macrophages. *Inflammation*, doi:10.1007/s10753-025-02304-6

(2025).

- 4 Zhao, J. *et al.* Tanshinone I specifically suppresses NLRP3 inflammasome activation by disrupting the association of NLRP3 and ASC. *Mol Med* **29**, 84, doi:10.1186/s10020-023-00671-0 (2023).
- 5 Sun, Z. *et al.* Acetaminophen-induced reduction of NIMA-related kinase 7 expression exacerbates acute liver injury. *JHEP reports : innovation in hepatology* **4**, 100545, doi:10.1016/j.jhepr.2022.100545 (2022).
- 6 Eisa, N. H., El-Sherbiny, M. & Abo El-Magd, N. F. Betulin alleviates cisplatin-induced hepatic injury in rats: Targeting apoptosis and Nek7-independent NLRP3 inflammasome pathways. *International immunopharmacology* **99**, 107925, doi:10.1016/j.intimp.2021.107925 (2021).
- 7 Schmacke, N. A. *et al.* IKKbeta primes inflammasome formation by recruiting NLRP3 to the trans-Golgi network. *Immunity* **55**, 2271-2284 e2277, doi:10.1016/j.immuni.2022.10.021 (2022).
- 8 Machtens, D. A., Bresch, I. P., Eberhage, J., Reubold, T. F. & Eschenburg, S. The Inflammasome Activity of NLRP3 Is Independent of NEK7 in HEK293 Cells Co-Expressing ASC. *International journal of molecular sciences* **23**, doi:10.3390/ijms231810269 (2022).
- 9 Zhang, J. *et al.* Fluorofenidone attenuates choline-deficient, l-amino acid-defined, high-fat diet-induced metabolic dysfunction-associated steatohepatitis in mice. *Sci Rep* **15**, 9863, doi:10.1038/s41598-025-94401-7 (2025).
- 10 Zhu, M. J. *et al.* Therapeutic role of *Prunella vulgaris* L. polysaccharides in non-alcoholic steatohepatitis and gut dysbiosis. *J Integr Med*, doi:10.1016/j.joim.2025.03.002 (2025).
- 11 Shi, H. *et al.* NLRP3 activation and mitosis are mutually exclusive events coordinated by NEK7, a new inflammasome component. *Nature immunology* **17**, 250-258, doi:10.1038/ni.3333 (2016).
- 12 Wang, Q. *et al.* Naringenin attenuates non-alcoholic fatty liver disease by down-regulating the NLRP3/NF-kappaB pathway in mice. *Br J Pharmacol* **177**, 1806-1821, doi:10.1111/bph.14938 (2020).
- 13 Mariathasan, S. *et al.* Cryopyrin activates the inflammasome in response to toxins and ATP. *Nature* **440**, 228-232, doi:10.1038/nature04515 (2006).
- 14 Simoni, C. *et al.* Liver fibrosis negatively impacts in vivo gene transfer to murine hepatocytes. *Nat Commun* **16**, 2119, doi:10.1038/s41467-025-57383-8 (2025).
- 15 Scholten, D., Trebicka, J., Liedtke, C. & Weiskirchen, R. The carbon tetrachloride model in mice. *Lab Anim* **49**, 4-11, doi:10.1177/0023677215571192 (2015).
- 16 Constandinou, C., Henderson, N. & Iredale, J. P. Modeling liver fibrosis in rodents. *Methods Mol Med* **117**, 237-250, doi:10.1385/1-59259-940-0:237 (2005).
- 17 Chen, S. *et al.* Ginsenoside Rh2 attenuates CDAHFD-induced liver fibrosis in mice by improving intestinal microbial composition and regulating LPS-mediated autophagy. *Phytomedicine* **101**, 154121, doi:10.1016/j.phymed.2022.154121 (2022).
- 18 Matsumoto, M. *et al.* An improved mouse model that rapidly develops fibrosis in non-alcoholic steatohepatitis. *Int J Exp Pathol* **94**, 93-103, doi:10.1111/iep.12008 (2013).

Response to Reviewer #2:

The manuscript by Sun et al. describes how NEK7 interacts with SDHB to regulate respiratory chain electron transport homeostasis, thereby mitigating liver fibrosis. This study, based on RNA sequencing of liver tissue from NEK7 hepatocyte-knockout mice, reveals that NEK7 deficiency impairs mitochondrial oxidative phosphorylation and downregulates the expression of genes associated with the mitochondrial inner membrane. The mitochondrial localization of NEK7 may be mediated by mitochondrial targeting signal peptides, and its absence appears to induce mitochondrial dysfunction. Mice with liver-specific NEK7 knockout exhibit spontaneous liver dysfunction and fibrosis. While the findings may interest researchers in the field, the precise role of NEK7 in mitochondria remains unclear, and the preliminary nature of many of the data limits the ability to draw definitive conclusions. Specific questions are as follows:

Response: We appreciate the reviewer's positive comments on our research work.

Major:

(1) NEK7 is known to play a key role in mitosis and the activation of the NLRP3 inflammasome ([doi: 10.1038/ni.3333](https://doi.org/10.1038/ni.3333), [doi: 10.1038/s41586-019-1295-z](https://doi.org/10.1038/s41586-019-1295-z)). However, in Fig. 1a, KEGG analysis of downregulated genes in NEK7 knockout mice did not reveal alterations in cell cycle or inflammation-related pathways. In contrast, Fig. 4 demonstrates a significant enhancement of inflammatory signaling in liver cells from NEK7 knockout mice. The authors are requested to clarify the rationale behind selecting downregulated genes for KEGG analysis and to explain why the analysis did not identify changes in cell cycle or inflammation-related pathways.

Response: Thanks for the Reviewer's valuable comments. Initially, we performed KEGG pathway analysis on the upregulated genes (Hom/Ctrl) and found that the significantly enriched pathways (Top 20) were primarily associated with inflammation and fibrosis (**see the Response Fig.1 below**). The NLRP3 inflammasome pathway (NOD-like receptor signaling pathway) was also significantly upregulated (ranked 104th, Q value < 0.0090, Rich Ratio = 0.2), which is contrary to previous reports suggesting that NEK7 positively regulates the NLRP3 inflammasome^{1,2}. Subsequently, we analyzed the downregulated genes (Hom/Ctrl) using KEGG pathway analysis (**Fig.1a**), and found the oxidative phosphorylation pathway was top ranked, which could explain the phenotype of liver injury (including the phenotypes in inflammation and fibrosis in NEK7 knockout mouse liver) due to loss of NEK7 in hepatocytes. However, no previous studies have reported the role of NEK7 in mitochondria. Therefore, we performed the series of investigations in this study and revealed the novel role of NEK7 in the mitochondria of hepatocytes.

NEK7 is also known to play a key role in mitosis³⁻⁵ and affects cell cycle by regulating cyclins, especially cyclin B1⁵. Here in the KEGG analysis of downregulated genes in NEK7 knockout mice, the cell cycle pathway was not significantly enriched (ranked 173rd, Q value = 1, Rich Ratio = 0.07), perhaps because only a few genes (9/123) involved in the cell cycle were affected, or the difference of status of liver pathology.

Response Fig.1 KEGG pathway analysis on the upregulated genes (Hom/Ctrl).

a, KEGG analysis of the upregulated genes (compared to the WT mice livers) in the Hom mice livers (Top20 pathways were displayed). N=3/group.

(2) In Figs. 1c and 1d, GAPDH is detected in both the mitochondrial and cytoplasmic fractions. As GAPDH is present in both compartments, its use as an internal reference protein for the mitochondrial fraction does not provide a reliable indication of the separation efficiency between the two fractions. It is recommended to use ACTB, a cytoskeletal protein that is absent from the mitochondrial fraction, as an internal reference protein for the cytoplasmic fraction ([doi: 10.1080/15548627](https://doi.org/10.1080/15548627)). This would better assess the purity of the mitochondrial and cytoplasmic preparations.

Response: Thanks for the Reviewer's valuable suggestions. We have changed the internal reference protein from GAPDH to β -actin (**Fig.1c and 1d**). At the beginning during the revision, we firstly tried to apply β -actin on the corresponding original membrane. The membrane in **Fig.1c** could work, but the membrane in **Fig.1d** couldn't work possibly because it had been stored too long. Therefore, we conducted a western blotting experiment using the same protein lysate again, and then changed all the bands (**Fig.1d**).

Fig.1 NEK7 is localized in mitochondria, mediated by its MTS peptides.

c,d, Western blotting analysis showing the NEK7 protein levels in mitochondria and cytoplasm isolated from AML-12 and HepG2 cells (n=3). Mito, mitochondria; Cyto, cytoplasm.

(3) In Fig. 1f, the expression level of NEK7- Δ MTS is notably lower than that of NEK7-WT, raising the possibility that the observed decrease in NEK7 mitochondrial co-localization may be due to reduced protein expression. Additionally, a significant difference in MitoTracker staining between the two groups could impact the accuracy of the co-localization quantification.

Response: Thanks for the important comments. We used the same quality of plasmids in both groups (NEK7- Δ MTS and NEK7-WT) during transfection and ensured the same concentration of dye and staining time during this experiment (including Flag and MitoTracker staining). To avoid bias, the images covering various fields of cells with fluorescence staining (including Flag and MitoTracker staining) were captured in both groups. Unfortunately, the previously presented image was not very presentative. Therefore, to better support our conclusions, we presented another representative image (NEK7- Δ MTS group in **Fig.1f**) with comparable fluorescence intensity from the images in NEK7- Δ MTS group.

Fig.1 NEK7 is localized in mitochondria, mediated by its MTS peptides.

f-g, Representative images (**f**) and the co-localization analysis (**g**) showing the different distribution pattern of NEK7^{WT}-flag (full-length NEK7) and NEK7^{ΔMTS}-flag (the predicted MTS sequences are deleted) in the mitochondria of HepG2 cells. N=3/group. Flag, green; MitoTracker, Red.

(4) Fig.3l,m: The binding of NEK7 and SDHB proteins has not been validated through reverse experimentation. It is recommended to introduce mutations at the predicted binding site and subsequently assess the direct interaction between the mutated proteins.

Response: Thanks for the Reviewer's valuable suggestions. To perform the reverse experiment, first the NEK7-mut plasmids with all the predicted binding sites (LYS74、ASN211、HIS209、GLU251 and TYR244) deleted were constructed, and then enough purified proteins (NEK7-mut) were obtained from the *e coli*.. Next, the SPR test using NEK7-mut and SDHB proteins were performed. This SPR result (**Supplementary Fig.4h**) indicated that the binding affinity of NEK7 and SDHB protein remarkably decreased (the K_D value increases from 0.349 μm to 4.75 μm) with the deletion of the binding sites in NEK7, which further confirms the direct interaction between NEK7 and SDHB. We thank the reviewer for this nice suggestion.

Supplementary Fig.4 Validation of the interaction of NEK7 and SDHB in AML-12 cells and HepG2 cells.

h, SPR analysis showing the binding affinity of NEK7-mut and SDHB protein.

(5) In Figs. 4o-q, liver-specific knockout of NEK7 triggers the activation of inflammation. However, other studies have reported that NEK7 promotes NLRP3-dependent cellular inflammatory responses ([doi: 10.1038/ni.3333](https://doi.org/10.1038/ni.3333)), and that NEK7 knockout alleviates chronic colitis in vivo ([doi: 10.1038/s41419-019-2157-1](https://doi.org/10.1038/s41419-019-2157-1)). The authors are encouraged to include measurements of NLRP3 expression levels in the liver of NEK7 liver-specific knockout mice and to provide an explanation for the seemingly bidirectional effects of NEK7 knockout on the inflammatory response.

Response: Thanks for the Reviewer's valuable suggestions. We performed additional experiments and found that the NLRP3 inflammasome was not significantly activated in the liver tissues and primary hepatocytes from the NEK7 knockout mice (**Supplementary Fig.10k-u**). The increased inflammation observed in the liver tissues of NEK7 knockout mice (**Fig. 4o-q**) may be resulted from hepatocyte damage caused by NEK7 deficiency-mediated mitochondrial dysfunction, which was confirmed by that hepatocyte NEK7 deficiency-related LX-2 activation and progression of liver fibrosis could be blocked by the downstream ROS scavenge and RET inhibitors (**Fig. 5**).

Previous studies have reported that in certain conditions, the assembly of the inflammasome mediated by the binding of NEK7 to NLRP3 is an important mechanism for its activation and promotion of inflammatory responses^{1,2,6}. However, in the field of liver injury or fibrosis, only the protein level of NEK7 or the pull-down level of NEK7 immunoprecipitated by NLRP3 were examined under disease conditions, no enough further evidence performed to demonstrate that the activation of the NLRP3 inflammasome is dependent on NEK7⁷⁻⁹.

Following your important comments, our additional experiments showed that neither overexpression nor deficiency of NEK7 in hepatocytes affected the activation of NLRP3 inflammasome (**Supplementary Fig.10**), indicating that NEK7 is not essential for NLRP3 activation in hepatocytes in the present experimental settings. Moreover, studies have also shown that in some pathological conditions, the activation of NLRP3 did not depend on NEK7¹⁰⁻¹².

Supplementary Fig.10 Effects of NEK7 on the activation of NLRP3 inflammasome determined in diverse models.

a-c, Western blotting analysis showing the protein levels of NLRP3 and Caspase 1 in the mouse livers of the NEK7-CDAHFD model. The quantification is in **(b,c)**. N=8/group.

d, Relative level of Caspase 1 activity in the livers. N=8/group.

e, Relative level of IL-1 β in the livers tested by ELISA. N=8/group.

f-h, Western blotting analysis showing the protein levels of NLRP3 and Caspase 1 in the mouse livers of the NEK7-AAV-CCl₄ model. The quantification is in **(g,h)**. N=6/group.

i, Relative level of Caspase 1 activity in the livers. N=6/group.
j, Relative level of IL-1 β in the livers tested by ELISA. N=6/group.
k-m, Western blotting analysis showing the protein levels of NLRP3 and Caspase 1 in the livers from the NEK7 knockout mice. The quantification is in (**l,m**). N=6/group.
n, Relative level of Caspase 1 activity in the livers. N=6/group.
o, Relative level of IL-1 β in the livers tested by ELISA. N=6/group.
p-r, Western blotting analysis showing the protein levels of NLRP3 and Caspase 1 in primary hepatocytes isolated from the WT and Hom mice. The quantification is in (**q,r**). N=6/group.
s, Relative level of IL-1 β in the primary hepatocytes (from WT, Het, Hom) tested by ELISA. N=5/group.
t, Relative level of Caspase 1 activity in the primary hepatocytes. The primary hepatocytes isolated from the WT and Het mice livers were treated with LPS (1 μ g /ml) and ATP (2.5 mM) for 24 h. N=3/group.
u, Relative level of IL-1 β in the medium of the primary hepatocytes tested by ELISA. The hepatocytes were treated as shown in (**t**). N=3/group.

Data are presented as mean \pm SE. N value indicates independent replicates (from distinct samples). Significant differences are analyzed using unpaired two-tailed *t*-test (**g-j,q,r**), and a one-way ANOVA with Benjamini multiple comparisons (**b-e,l-o,s-u**).

(6) NOD-like receptor protein 3 (NLRP3) inflammasome activation occurs in Non-alcoholic fatty liver disease (NAFLD). Previous studies have demonstrated that inhibiting NLRP3 inflammasome activation reduces inflammatory recruitment and liver fibrosis in mouse models of steatohepatitis ([doi: 10.1016/j.jhep.2017.01.022](https://doi.org/10.1016/j.jhep.2017.01.022)). However, NEK7 promotes NLRP3 assembly and activation, and the authors' overexpression of NEK7 in NAFLD mice appeared to alleviate liver fibrosis. The authors are encouraged to supplement their study with an assessment of inflammasome levels following NEK7 overexpression in NAFLD mice and to provide an explanation for the dual effects of NEK7 in the NAFLD model.

Response: Thanks for the Reviewer's valuable suggestions. As you said, inflammasome activation plays an important role in the progression of NAFLD, with substantial evidence in the studies, including the expression and assembly of its components, increased caspase 1 activity, and elevated secretion of IL-1 β and IL-18¹³⁻¹⁵. Previous studies showed that in certain pathological conditions or cells, especially in BMDMs, NEK7 binds to NLRP3, promoting its assembly and activation^{1,2,6}. However, in NAFLD and other liver injury models, only the levels of NEK7 protein and its pulled-down levels by NLRP3 have been examined, lacking further evidence to demonstrate that NEK7 is indispensable for the inflammasome activation in injured livers⁷⁻⁹.

We performed additional experiments showing that overexpression of NEK7 did not affect NLRP3 inflammasome activation in the NAFLD model induced by CDAHFD (**Supplementary Fig.10a-e**). Moreover, compared with the wild-type cells, the activation of the inflammasome in primary hepatocytes induced by combination of LPS and ATP was completely unaffected

with NEK7 knockdown (**Supplementary Fig.10 t,u**), suggesting that NLRP3 inflammasome activation in hepatocytes may not depend on NEK7. Additionally, studies have reported other mechanisms of NLRP3 activation that are independent of NEK7¹⁰⁻¹². In this study, overexpression of NEK7 improves liver fibrosis mainly by enhancing mitochondrial function, thereby reducing hepatocyte damage and subsequent liver fibrosis.

Supplementary Fig.10 Effects of NEK7 on the activation of NLRP3 inflammasome determined in diverse models.

a-c, Western blotting analysis showing the protein level of NLRP3 and Caspase 1 in the mouse livers of the NEK7-CDAHFD model. The quantification is in **(b,c)**. N=8/group.

d, Relative level of Caspase 1 activity in the livers. N=8/group.

e, Relative level of IL-1 β in the livers tested by ELISA. N=8/group.

t, Relative level of Caspase 1 activity in the primary hepatocytes. The primary hepatocytes isolated from the WT and Het mice livers were treated with LPS (1 μ g/ml) and ATP (2.5 mM) for 24 h. N=3/group.

u, Relative level of IL-1 β in the medium of the primary hepatocytes tested by ELISA. The hepatocytes were treated as shown in **(t)**. N=3/group.

Data are presented as mean \pm SE. N value indicates independent replicates (from distinct samples). Significant differences are analyzed using a one-way ANOVA with Benjamini multiple comparisons (**b-e,t-u**).

Minor:

(1) Line130: The expression 'inhibition of NEK7 by siRNAs' is inappropriate.

Response: Thanks for the Reviewer's comments. We rewrote this expression as 'with inhibition of NEK7' in the revised manuscript.

(2) Line152: The abbreviation TMRE should appear at Line139, Fig.2h.

Response: Thanks for the Reviewer's nice comment. The position of the abbreviation TMRE has been adjusted in the revised manuscript following the Reviewer's suggestion.

(3) Line148 has already shown heterozygous (Het), so Line233 does not need further explanation.

Response: Thanks for the Reviewer's nice suggestions. Following the Reviewer's suggestion, we rewrote the corresponding sentence in the revised manuscript.

References

- 1 Shi, H. *et al.* NLRP3 activation and mitosis are mutually exclusive events coordinated by NEK7, a new inflammasome component. *Nature immunology* **17**, 250-258, doi:10.1038/ni.3333 (2016).
- 2 Sharif, H. *et al.* Structural mechanism for NEK7-licensed activation of NLRP3 inflammasome. *Nature* **570**, 338-343, doi:10.1038/s41586-019-1295-z (2019).
- 3 Sun, Z., Gong, W., Zhang, Y. & Jia, Z. Physiological and Pathological Roles of Mammalian NEK7. *Frontiers in physiology* **11**, 606996, doi:10.3389/fphys.2020.606996 (2020).
- 4 Yissachar, N., Salem, H., Tennenbaum, T. & Motro, B. Nek7 kinase is enriched at the centrosome, and is required for proper spindle assembly and mitotic progression. *FEBS letters* **580**, 6489-6495, doi:10.1016/j.febslet.2006.10.069 (2006).
- 5 Sun, Z. *et al.* Acetaminophen-induced reduction of NIMA-related kinase 7 expression exacerbates acute liver injury. *JHEP reports : innovation in hepatology* **4**, 100545, doi:10.1016/j.jhepr.2022.100545 (2022).
- 6 Chen, X. *et al.* NEK7 interacts with NLRP3 to modulate the pyroptosis in inflammatory bowel disease via NF-kappaB signaling. *Cell Death Dis* **10**, 906, doi:10.1038/s41419-019-2157-1 (2019).
- 7 Sheng, M. *et al.* CD47-Mediated Hedgehog/SMO/GLI1 Signaling Promotes Mesenchymal Stem Cell Immunomodulation in Mouse Liver Inflammation. *Hepatology* **74**, 1560-1577, doi:10.1002/hep.31831 (2021).
- 8 Liu, J. *et al.* CLICs Inhibitor IAA94 Alleviates Inflammation and Injury in Septic Liver by Preventing Pyroptosis in Macrophages. *Inflammation*, doi:10.1007/s10753-025-02304-6 (2025).

- 9 Zhao, J. *et al.* Tanshinone I specifically suppresses NLRP3 inflammasome activation by disrupting the association of NLRP3 and ASC. *Mol Med* **29**, 84, doi:10.1186/s10020-023-00671-0 (2023).
- 10 Eisa, N. H., El-Sherbiny, M. & Abo El-Magd, N. F. Betulin alleviates cisplatin-induced hepatic injury in rats: Targeting apoptosis and Nek7-independent NLRP3 inflammasome pathways. *International immunopharmacology* **99**, 107925, doi:10.1016/j.intimp.2021.107925 (2021).
- 11 Schmacke, N. A. *et al.* IKKbeta primes inflammasome formation by recruiting NLRP3 to the trans-Golgi network. *Immunity* **55**, 2271-2284 e2277, doi:10.1016/j.immuni.2022.10.021 (2022).
- 12 Machtens, D. A., Bresch, I. P., Eberhage, J., Reubold, T. F. & Eschenburg, S. The Inflammasome Activity of NLRP3 Is Independent of NEK7 in HEK293 Cells Co-Expressing ASC. *International journal of molecular sciences* **23**, doi:10.3390/ijms231810269 (2022).
- 13 de Carvalho Ribeiro, M. & Szabo, G. Role of the Inflammasome in Liver Disease. *Annu Rev Pathol* **17**, 345-365, doi:10.1146/annurev-pathmechdis-032521-102529 (2022).
- 14 Mridha, A. R. *et al.* NLRP3 inflammasome blockade reduces liver inflammation and fibrosis in experimental NASH in mice. *Journal of hepatology* **66**, 1037-1046, doi:10.1016/j.jhep.2017.01.022 (2017).
- 15 Zhu, M. J. *et al.* Therapeutic role of *Prunella vulgaris* L. polysaccharides in non-alcoholic steatohepatitis and gut dysbiosis. *J Integr Med*, doi:10.1016/j.joim.2025.03.002 (2025).

Response to Reviewer 3#:

I have reviewed the manuscript titled "NEK7 couples SDHB to orchestrate respiratory chain electron transport homeostasis that impedes liver fibrosis." The study explores a novel role for NEK7 in mitochondrial function by demonstrating its interaction with SDHB and its potential regulatory role in Complex II activity. The authors suggest that NEK7 deletion leads to mitochondrial dysfunction, increased ROS, and enhanced liver fibrosis, while NEK7 overexpression mitigates these effects.

This study presents an interesting perspective on the role of NEK7 in mitochondrial metabolism and liver fibrosis. However, there are several key concerns regarding the mechanistic depth of the findings, data interpretation, and experimental validation. While the proposed model is intriguing, additional evidence is needed to support the claim that NEK7 functions as a regulator of Complex II activity rather than as a structural scaffold. Furthermore, the specificity of DMM's effect in rescuing NEK7 deficiency has not been fully validated. Given these points, I believe that further experimental refinement is necessary before this work can be considered for publication in Nature Communications. Below, I outline major areas where additional data and clarifications would strengthen the manuscript.

Response: We appreciate for the important suggestions. Following the valuable comments, we performed additional experiments to address the concerns and believe that the quality of

the manuscript was significantly improved after the revision. The detailed responses were presented below.

Major Concerns

1. Insufficient Evidence Linking NEK7-SDHB Interaction to Complex II Functional Regulation

The central claim of the manuscript is that NEK7 directly modulates Complex II function through its interaction with SDHB. However, the authors do not provide sufficient experimental evidence to support this conclusion. While Co-IP and Mass Spectrometry data suggest an interaction between NEK7 and SDHB, this does not necessarily indicate a functional role.

- The study lacks direct measurements of Complex II enzymatic activity (e.g., SDH activity assay, succinate oxidation rate) in NEK7-deficient and overexpressing cells.
- It remains unclear whether NEK7 regulates Complex II enzymatic function or merely stabilizes SDHB as a scaffold protein.
- Post-translational modifications (PTMs) of SDHB (e.g., phosphorylation or ubiquitination) that could be influenced by NEK7 were not investigated.
- Blue Native PAGE or protein stability assays should be performed to confirm whether NEK7 affects Complex II assembly and integrity.

Conclusion: Additional functional assays would help establish whether NEK7 directly regulates Complex II enzymatic activity or if its role is primarily structural.

Response: Thanks for the Reviewer's valuable suggestions. In this study, we showed that NEK7 deficiency disrupted the activity of mitochondrial complex II (**Fig.3n**). We also examined the effect of NEK7 overexpression on the activity of complex II, and no significant difference was observed (under normal condition, the amount of NEK7 in the mitochondria of hepatocytes could be enough to couple SDHB), which was further confirmed by additional repeated experiments again during the revision (**Supplementary Fig.4i**). At the same time, in vitro, the abnormal complex II activity caused by CCl₄ and the generation of ROS resulting from increased RET caused by succinate-induced complex II activity could all be rescued by NEK7 overexpression (**Supplementary Fig.9g and Fig.3s**).

Following the reviewer's suggestions, we also performed Blue Native Page assay, and found that neither NEK7 overexpression nor deficiency affected the complex II assembly and integrity, compared to the control (**Supplementary Fig.4f,g**). In addition, we did not find any published works suggested the precise role of post-translational modifications of SDHB in the activity of complex II, here we also investigated whether the post-translational modifications

of SDHB (including phosphorylation or ubiquitination) were potentially affected by NEK7. We first performed CO-IP using an anti-SDHB antibody, followed by the western blotting using pan P-Ser/Thr and pan Poly-Ub antibodies, due to the absence of specific SDHB phosphorylation and ubiquitination antibodies. We found that the gray level of the bands near the SDHB protein (28 KDa) did not show obvious difference between Vec and NEK7-overexpression groups regardless of ubiquitination or phosphorylation (**Response Fig.2**).

Taken together, these results further validate that NEK7 could keep the spatial conformation stability of complex II, thus maintains its activity, through binding with SDHB. We thank the reviewer's comments for better addressing the concept.

Supplementary Fig.4 Validation of the interaction of NEK7 and SDHB in AML-12 cells and HepG2 cells.

f,g, Blue Native PAGE analysis of the mitochondrial fractions isolated from the HepG2 cells transfected with NC/si-NEK7 or Vec/NEK7 for 48h. Complex II and complex IV was probed by anti-SDHB and anti-COX1 respectively, complex IV, as the control.

i, Mitochondrial complex II activity in the HepG2 cells transfected with NEK7 or Vec for 48 h.

Data are presented as mean \pm SE. N=3/group. Significant differences are calculated using unpaired two-tailed *t*-test (i).

Response Fig.2 Role of NEK7 in SDHB post-translational modifications in HepG2 cells.

a, Western blotting analysis following by CO-IP with an SDHB antibody showing the level of pan phosphorylation (P-Ser/Thr) and ubiquitination (Poly-Ub).

2. Lack of Evidence Supporting the NEK7-Specific Effect of Dimethyl Malonate (DMM) (Figure 5)

The authors propose that Dimethyl Malonate (DMM) rescues mitochondrial dysfunction and fibrosis in NEK7-deficient cells. However, it is unclear whether this effect is NEK7-specific or a general metabolic phenomenon related to SDH inhibition.

- DMM is a well-known SDH inhibitor and could exert protective effects through general metabolic modulation rather than NEK7-specific pathways.
- Control experiments using wild-type cells treated with DMM are necessary to determine whether DMM's effect is specific to NEK7 deletion.
- The authors do not show whether NEK7 regulates SDH activity directly. If DMM rescues fibrosis independently of NEK7 status, the authors' conclusions about NEK7's role are significantly weakened.

Conclusion: Further experiments are needed to confirm whether DMM's effects are directly linked to NEK7's function rather than being a general metabolic effect.

Response: Thank the Reviewer for this comment. Dimethyl Malonate (DMM) is a prodrug of malonate inhibiting ROS production from RET through binding to the carboxylate binding site in complex II¹⁻³. Reverse electron transfer can be triggered by various causes, accompanied by increased mitochondrial complex II activity and membrane potential, as well as uncoupled oxidative phosphorylation, ultimately leading to the generation of a large amount of ROS²⁻⁴. In this study, we found that all the phenotypes of NEK7 deficiency in hepatocytes, such as increased mitochondrial complex II activity and membrane potential, uncoupled oxidative phosphorylation, a large amount of ROS production, indicated the occurrence of RET (**Fig.2 and Fig.3n**). Furthermore, overexpression of NEK7 could rescue the abnormal activity of complex II disrupted by CCl₄ and decreased the level of ROS resulted from the RET induced by succinate (**Supplementary Fig.9g and Fig3s**). Following with NEK7-inhibition, DMM treatment could inhibit ROS generation, hepatocyte injury and activation of hepatic stellate cells caused by NEK7 deletion, which confirmed that RET could be the key mechanism in hepatocyte damage due to NEK7-deficiency. Following your important suggestion, we performed additional experiments and the data showed that DMM treatment alone did not significantly affected ROS level, membrane potential, complex II activity in HepG2 cells, and DMM-conditioned medium of HepG2 cells also did not alter the α -SMA level in LX-2 cells (**Supplementary Fig.9**). Therefore, all these evidences indicated that NEK7 played an important role in maintaining complex II activity, and the effects of DMM in vivo and in vitro confirmed that RET was the key mechanism downstream of NEK7-deficiency in hepatocyte damage.

Supplementary Fig.9

Supplementary Fig.9 Role of DMM in mitochondrial function and complex II activity in HepG2 cells.

a, Flow cytometry detected mitochondrial ROS levels (MitoSox Red) in HepG2 cells. The HepG2 cells are treated with 10 μ M DMM or vehi (Ctrl) for 24 h.

b, Flow cytometry detected mitochondrial membrane potential (TMRE) in control and DMM-treated HepG2 cells.

c, Activity of mitochondrial complex II in the HepG2 cells treated with 10 μ M DMM or vehi (Ctrl) for 24 h.

d,e, Representative images (**e**) of α -SMA staining in LX-2 cells (scale bar, 20 μ m), the LX-2 cells are treated with the corresponding conditional medium of the HepG2 cells in (**a**). Relative fluorescence intensity is displayed in (**d**).

f, Representative TEM images depicting the ultrastructure of the mitochondria in hepatocytes from the Het+CCl₄ and Het+CCl₄+DMM mice (scale bar, 5 μ m).

g, Mitochondrial complex II activity in the HepG2 cells. The control (Vec) and NEK7-overexpressed HepG2 cells are treated with vehicle (Vehi) or CCl₄ (5 mM) for 24 h.

Data are presented as mean \pm SE. N=3/group. Significant differences are calculated using unpaired two-tailed *t*-test (**a-d**), and a one-way ANOVA with Benjamini multiple comparisons (**g**).

3. NEK7 Overexpression and its Functional Connection to Complex II Activity

- The authors provide phenotypic evidence that NEK7 overexpression reduces fibrosis, but they do not establish whether this is directly linked to Complex II function.
- If NEK7 regulates Complex II, its overexpression should enhance Complex II activity (measurable through SDH assays), but such data are missing.
- NEK7's potential role in mitochondrial structure or stability was not explored, leaving open the possibility that its effect on fibrosis is independent of Complex II activity.

Conclusion: Additional biochemical and structural analyses would help clarify the connection between NEK7 overexpression and Complex II function.

Response: Thanks for the reviewer's important suggestion. As mentioned in the above response, NEK7 deficiency led to increased activity of complex II (**Fig.3n**). Together with other data in **Fig.1-3**, we proposed that NEK7 deficiency in hepatocytes led to RET resulting in more mtROS generation and release, which was considered to be the key link in causing spontaneous liver fibrosis of the NEK7 knockout mice and also in aggravating CCl₄-induced liver fibrosis. Moreover, inhibiting the RET (by DMM, or Rot, or Met) or scavenging ROS (by Tempo) all alleviated the phenotype resulting from NEK7 deficiency both in vitro and in vivo.

Furthermore, although NEK7 overexpression did not significantly affect the post-translational modifications of SDHB and the level of complex II as well as the complex II activity under physiological condition (**Response Fig.2, Supplementary Fig.4g,i**), pre-overexpression of NEK7 could rescue the abnormal activity of complex II disrupted by CCl₄ and decreased the level of ROS resulted from the RET induced by succinate (**Supplementary Fig.9g and Fig.3s**), which is therefore considered the molecular mechanism for attenuating CCl₄- and CDAHFD- induced liver fibrosis in vivo. These data suggested that abundant NEK7 is crucial for complex II to maintain homeostasis and resist damage.

Abnormality of mitochondrial complexes can lead to structural abnormalities in mitochondria⁵⁻⁸. Therefore, we speculated that abnormalities of mitochondrial structure (including decreased cristae density, mitochondrial swelling, and vacuolization) due to loss of NEK7, as shown in the TEM images (**Fig.2a,f**), may be resulted from the abnormal complex II activities and the disordered electron transfer. Additionally, NEK7 overexpression by NEK7-AAV (**Fig.7n**) or inhibiting the downstream RET by DMM (**Supplementary Fig.9f**) significantly restored mitochondrial morphology and structure. These results again supported that NEK7 was linked to complex II function.

4. Lack of Clinical Relevance and Validation in Human Samples

To support the significance of their findings, the authors should consider validating their results in liver fibrosis patient samples.

- Is NEK7 downregulated in human liver fibrosis patients?
- Does NEK7 expression correlate with mitochondrial dysfunction markers in human liver samples?
- If NEK7-SDHB signaling is a conserved fibrosis-regulating pathway, it should be detectable in human samples.

Conclusion: Incorporating human sample validation would enhance the translational relevance of the study.

Response: Thanks for the Reviewer's suggestion. We apologized for not being able to obtain adequate samples of liver fibrosis in the short term. As you know, we are from a children's hospital (Children's Hospital of Nanjing Medical University). In a children's hospital, we have no biopsy samples of chronic liver fibrosis. However, we have liver samples from hepatoblastoma children and examined the subcellular localization of NEK7 in hepatocytes using the paraffin-embedded sections derived from normal liver tissues adjacent to the hepatoblastoma in patients from our hospital, which showed a high localization of NEK7 in COXIV-labeled mitochondria (**Supplementary Fig.1f**). In addition, we extracted mitochondria and cytoplasm fractions from normal liver tissue adjacent to the tumor and detected the level of NEK7. We found that there is also a high abundance of NEK7 in the human hepatic mitochondria (**Supplementary Fig.1g**), consistent with the results from the hepatocyte cell lines and mouse liver tissues in the manuscript, further confirming the distribution of NEK7 in hepatocyte mitochondria. Due to the high conservation of NEK7 between human and mouse, combined with the important role of NEK7 in mitochondria of human hepatocyte lines investigated in this study, it suggests that NEK7-orchestrated mitochondrial complex II is crucial for the maintenance of mitochondrial homeostasis and normal hepatocyte function. Undoubtedly, further exploration using human chronic liver disease samples would bring more benefit for clinical translation, which is also the limitation of this study. We would like to make the collaboration with the liver researchers from the adult hospital to examine more using the fibrotic liver biopsy samples.

Supplementary Fig.1 Co-localization analysis of NEK7 and mitochondria.

f, Representative images of the co-localization staining of NEK7 (Green) and mitochondria (COXIV, Red) in human liver tissues. Scale bar, 20 μ m. N=5 (patient 1#-5#, shown in **Supplementary Table 1**).

g, Western blotting analysis showing the NEK7 protein levels in mitochondria and cytoplasm isolated from human liver tissues (N=3, patient 6#-8#, shown in **Supplementary Table 1**). Mito, mitochondria; Cyto, cytoplasm.

5. Lack of Liver-Specificity in pcDNA Injection and Potential Immune Effects

- The authors used pcDNA injection to overexpress NEK7, but this vector does not have liver specificity and may lead to NEK7 expression in immune cells as well.
- NEK7 is known to promote inflammasome activation in immune cells, which could result in increased immune activation rather than a direct antifibrotic effect in hepatocytes (DOI: [10.4049/jimmunol.208.Supp.52.02](https://doi.org/10.4049/jimmunol.208.Supp.52.02), [10.1126/sciimmunol.adl2993](https://doi.org/10.1126/sciimmunol.adl2993)).
- To confirm that NEK7's effect is liver-specific, experiments should be conducted using a liver-specific promoter (e.g., Albumin promoter, AAV8/9-based delivery) to restrict expression to hepatocytes.

Conclusion: The use of pcDNA raises concerns that NEK7 overexpression may have influenced immune cells rather than acting specifically in hepatocytes. Liver-specific promoters should be used to verify that the observed effects are hepatocyte-driven.

Response: We agree with the Reviewer's suggestions. To confirm that NEK7's effect is liver-specific, we constructed AAV8-TBG-NEK7 (NEK7-AAV for short), which overexpressed NEK7 specifically and efficiently in hepatocytes (**Fig. 7a, d**). Four-weeks after NEK7-AAV injection, the mice were administered with CCl₄ or Vehi twice a week for 6-7 weeks. Consistent with the protection of NEK7-plasmids tail vein injection, impaired liver function, fibrotic deposition, activation of the hepatic stellate cells, and oxidative damage, which were induced by CCl₄, were all markedly alleviated in the mice with NEK7-AAV treatment (**Fig. 7e-**

m, o-v). Additionally, we also observed that the damaged mitochondria by CCl₄ were rescued obviously, manifested in the recovery of mitochondrial morphology and cristae density, and the reduction of vacuolar degeneration (Fig. 7n).

Fig.7 NEK7 overexpression specific in hepatocytes by AAV8-TBG-NEK7 alleviates CCl₄-induced liver fibrosis

a,b, Western blotting analysis showing NEK7 expression in the livers of the mice injected with NEK7-AAV (AAV8-TBG-NEK7-EGFP) or Ctrl-AAV (empty vector). The quantification is in (b).

c,d, Representative images and quantification of NEK7 IF staining (scale bar, 20 μ m).

e,f, Levels of serum ALT and AST in the mice with diverse treatments (see Methods).

g, Western blotting analysis of fibrotic protein levels (including Collagen I and α -SMA) in the livers of the mice. The quantifications are displayed in (h,i).

j-m, Relative mRNA levels of *Collagen I*, *Collagen III*, *MMP-2*, and *MCP-1* in the livers of the mice with diverse treatment (see Methods).

n, Representative TEM images depicting the ultrastructure of the mitochondria in hepatocytes from each

group (scale bar, 500 nm), n=4/group.

o-t, Representative images and quantification of α -SMA IF (scale bar, 20 μ m), Sirius Red (scale bar, 100 μ m), and Masson staining (scale bar, 100 μ m).

u,v, Representative images and quantification of 4-HNE IHC staining (scale bar, 100 μ m).

Data are presented as mean \pm SE. N value indicates independent replicates (from distinct samples). Significant differences are analyzed using unpaired two-tailed *t*-test (**b,d**, n=6/group), and a one-way ANOVA with Benjamini multiple comparisons (**e,f,h-m,p,r,t,v**, n=6/group).

Given the lack of direct evidence for NEK7-mediated Complex II regulation, the absence of functional validation in overexpression models, the potential confounding effects of DMM, the lack of liver specificity in pcDNA injection, and the lack of clinical validation, the current manuscript does not meet the high standards required for publication in Nature Communications. I encourage the authors to refine their mechanistic insights and provide stronger experimental validation before resubmitting to a high-impact journal.

Response: We appreciate the valuable comments for further strengthening the quality of this research work. Following your important suggestions, we performed additional experiments and provided comprehensive revision on the manuscript. We think these revisions significantly enhanced the quality of the manuscript and thank the valuable comments from the reviewer again!

References

- 1 Mills, E. L. *et al.* Succinate Dehydrogenase Supports Metabolic Repurposing of Mitochondria to Drive Inflammatory Macrophages. *Cell* **167**, 457-470 e413, doi:10.1016/j.cell.2016.08.064 (2016).
- 2 Drose, S. Differential effects of complex II on mitochondrial ROS production and their relation to cardioprotective pre- and postconditioning. *Biochimica et biophysica acta* **1827**, 578-587, doi:10.1016/j.bbabi.2013.01.004 (2013).
- 3 Prag, H. A. *et al.* Ester Prodrugs of Malonate with Enhanced Intracellular Delivery Protect Against Cardiac Ischemia-Reperfusion Injury In Vivo. *Cardiovascular drugs and therapy* **36**, 1-13, doi:10.1007/s10557-020-07033-6 (2022).
- 4 Quinlan, C. L. *et al.* Mitochondrial complex II can generate reactive oxygen species at high rates in both the forward and reverse reactions. *The Journal of biological chemistry* **287**, 27255-27264, doi:10.1074/jbc.M112.374629 (2012).
- 5 Vercellino, I. & Sazanov, L. A. The assembly, regulation and function of the mitochondrial respiratory chain. *Nature reviews. Molecular cell biology* **23**, 141-161, doi:10.1038/s41580-021-00415-0 (2022).
- 6 Cao, K. *et al.* Assembly of mitochondrial succinate dehydrogenase in human health and disease. *Free radical biology & medicine* **207**, 247-259, doi:10.1016/j.freeradbiomed.2023.07.023 (2023).

- 7 Stefanatos, R. *et al.* Developmental mitochondrial complex I activity determines lifespan. *EMBO reports* **26**, 1957-1983, doi:10.1038/s44319-025-00416-6 (2025).
- 8 Bahety, D., Boke, E. & Rodriguez-Nuevo, A. Mitochondrial morphology, distribution and activity during oocyte development. *Trends Endocrinol Metab* **35**, 902-917, doi:10.1016/j.tem.2024.03.002 (2024).

Response to Reviewer #4:

Dear Authors, Thank you for a significant and important paper in the field of liver fibrosis. This reviewer find the work on NEK7 important.

Response: We appreciate the reviewer's positive comments of our research work.

This reviewer have a couple of improvements needed to recommend a publication - not enough detail is included:

1.Line 535-536: The description of the proteomics analysis is insufficient in order for others to reproduce the experiments. This must be corrected according to the MIAPE/HUPO guidelines. See the following link: (<https://www.psidev.info/miape>), <https://pubmed.ncbi.nlm.nih.gov/17687369/>.

Minimum requirements are: Information on sample preparation protocol, chromatographic separation (column brand, polarity, gradient, time, solvents), mass spectrometry settings and software used. It can be very short with referring to literature method, but the above information must be included.

Example of level of details of proteomics for liver disease can be viewed in papers like: <https://pubmed.ncbi.nlm.nih.gov/35654907/>

Response: Thanks for the Reviewer's valuable suggestions. Following the Reviewer's suggestions, we provided more detailed information on sample preparation protocol, chromatographic separation (column brand, polarity, gradient, time, solvents), mass spectrometry settings and software used in the **Methods**.

2. Finally, at least a peak list of all of the 68 proteins pulldown by the CO-IP NEK7 must be in a supplementary file.

Line 540: From above – a peak list of the 68 proteins must be included in supplementary file.

Response: Thanks for the Reviewer's valuable suggestions. We provided the list of the 68 proteins in **Supplementary Data 1** in the revised manuscript.

3. Line 551-552: Details on digestion, sample processing, metabolite extraction and UHPLC analysis (column, gradient) is missing. Needed in order for others to reproduce experiments.

Response: Thanks for the Reviewer's valuable suggestions. We provided the details on the digestion, sample processing, metabolite extraction and analysis in the **Methods** (the sample processing, metabolite extraction and analysis were performed by BGI Genomics).

Response to the Reviewers:

Response to Reviewer #1:

The authors made a great job in preparing the revised version of the manuscript, which has significantly improved its quality. However, this reviewer is still not convinced of using cell lines instead of primary hepatocytes from mice, which are readily available after disaggregation of the liver tissue. This concern is related to comment number 4. This is my original comment: Surprisingly, in Figure 1, once NEK7 has been linked to downregulation of oxidative phosphorylation in NEK7 hepatocyte-specific KO mice, the authors jump into the human describing the protein expression and the localization of NEK7 in a human hepatocyte cell line derived from a patient with hepatoblastoma. For consistency, it is necessary to perform these analyses in murine hepatocytes. This is also true for Figure 2, in which the role of NEK7 in the mitochondria is mainly explored in the human cell line and then confirmed in primary mouse hepatocytes, thus following a nonlogical pattern of data presentation. This is also true for Figure 3.

It would be necessary to show data in primary murine hepatocytes instead of using AML-12 cells, unless the authors justify the technical reasons why they are not using the primary cell approach.

Response: Thanks for the valuable comments. Following the suggestions, we performed additional experiments in mouse primary hepatocytes (shown in the following details and the corresponding changes in the revised manuscript and figures).

In **Supplementary Fig.1**, we detected the level of NEK7 in the mitochondria and cytoplasm fractions from mouse primary hepatocytes (**Supplementary Fig.1f**), and performed co-localization staining of NEK7 and MitoTracker in mouse primary hepatocytes (**Supplementary Fig.1a**).

In **Supplementary Fig.4**, we performed additional CO-IP experiments (**Supplementary Fig.4a**) and co-localization staining of NEK7 and SDHB (**Supplementary Fig.4b**) in mouse primary hepatocytes.

All these results from the mouse primary hepatocytes were consistent with those from the cell lines (including AML-12, HepG2) and liver tissues, further supporting our conclusion

Supplementary Fig.1 Co-localization analysis of NEK7 and mitochondria
a, Representative images acquired by laser confocal microscopy depicting NEK7 distribution (Green) in the mitochondria (Red, MitoTracker) of mouse primary hepatocytes. Scale bar, 10 μ m. Pearson's Rr and Overlap R analyzed by the ImageJ plugin are 0.648 and 0.767, indicating a high co-localization of NEK7 with mitochondria. N=3.
f, Western blotting analysis showing the NEK7 protein levels in mitochondria and cytoplasm isolated from mouse primary hepatocytes. Mito, mitochondria; Cyto, cytoplasm.

Supplementary Fig.4 Validation of the interaction of NEK7 and SDHB in hepatocytes.

a, Western blotting analysis showing the interaction of NEK7 and SDHB determined by co-immunoprecipitation with IgG and NEK7 antibody (both in AML-12 cells and mouse primary hepatocytes).

b, Representative immunofluorescence images and the co-localization analysis show NEK7 and SDHB co-localization in mouse primary hepatocytes. Scale bar, 10 μ m. N=3.

Response to Reviewer #2:

The author has answered all my questions. I have no further questions for the author to respond to. I recommend that it be published in Nature Communications.

Response: We sincerely thank the Reviewer for the valuable suggestions that substantially strengthened and refined this study.

Response to Reviewer #3:

Comments to the Authors

This manuscript investigates the role of NEK7 in mitochondrial electron transport, proposing that NEK7 stabilizes Complex II via binding to SDHB, thereby suppressing reverse electron transport (RET) and reducing ROS overproduction, which in turn alleviates liver fibrosis. The authors present both in vitro and in vivo data supporting that NEK7 overexpression can mitigate liver fibrosis induced by CCl₄ and CDAHFD, while NEK7 deficiency worsens it. This is an original and potentially impactful study, highlighting a new mitochondrial regulatory mechanism.

However, I have significant concerns about the translational and mechanistic validity of the study that, in my view, preclude publication at this stage.

1. The central hypothesis is that boosting NEK7 could be therapeutic for liver fibrosis. However, the authors do not show whether NEK7 is actually downregulated in human fibrotic liver samples, or in standard mouse models of liver fibrosis, prior to their interventions. Without this information, it is unclear whether NEK7 represents a physiologically relevant therapeutic target. If NEK7 is not reduced during fibrosis, enhancing its expression may have limited translational value.

Response: Thank the Reviewer for the valuable comments. In this revised version of the manuscript, additional experiments were performed to investigate the level of NEK7 in the liver tissues and mitochondria from fibrotic mice and cell models. The expression of NEK7 was significantly reduced in the total and mitochondrial lysates from the mouse fibrotic livers induced by CCl₄ (**Supplementary Fig.10a-f**), as well as in the total and mitochondrial lysates of primary hepatocytes extracted from the fibrotic livers (**Supplementary Fig.10g-j**). In addition, NEK7 was also significantly downregulated in the mitochondria of the mouse fibrotic livers induced by CDAHFD (**Supplementary Fig.10k,l**). Furthermore, we found that the level of NEK7 in the fibrotic livers of NAFLD patients (fibrosis stage:4) showed a decrease trend with two tailed t-test (one tailed T-test showed significant decrement, GEO dataset GSE135251) (**Supplementary Fig.10m**). In human-derived HepG2 cells treated with CCl₄ and free fatty acids (FFA, palmitic acid/ oleate), the expression of NEK7 both in total and mitochondrial lysates was also significantly downregulated (**Supplementary Fig.10n-u**). Given the critical role of NEK7 in maintaining normal mitochondrial homeostasis in hepatocytes, the reduction of NEK7 could be detrimental in the occurrence and progression of liver fibrosis, better highlighting that NEK7 may be a promising therapeutic target for the treatment of chronic liver fibrosis. We do thank the reviewer's important comment on this point.

Supplementary Fig.10

Supplementary Fig.10 Investigation of the level of NEK7 in hepatocytes and fibrotic livers.

a,b, Representative images and quantification of NEK7 IF staining (scale bar, 20 μm).

c-f, Western blotting analysis showing the protein levels of NEK7 in the mouse livers of the CCl₄ model. Total, total lysates (**c,d**); Mito, mitochondrial lysates (**e,f**). The quantification is in (**d,f**). N=6/group.

g-j, Western blotting analysis showing the protein levels of NEK7 in the mouse primary hepatocytes of the CCl₄ model. Total, total lysates (**g,h**); Mito, mitochondrial lysates (**i,j**). The quantification is in (**h,j**). N=6/group.

k,l, Western blotting analysis showing the protein levels of NEK7 in mitochondria from the mouse livers of the CDAHFD model. N=6/group.

m, Relative TPM level of NEK7 in the livers from the control and NAFLD patients, from public GEO dataset GSE135251. 8 of control (fibrosis stage: 0) and 14 of NAFLD patients (fibrosis stage: 4) were included for analysis.

n-q, Western blotting analysis showing the protein levels of NEK7 in total (**n,o**) and mitochondrial (**p,q**) lysates of the HepG2 cells treated with CCl₄ (5 mM) and the Vehicle

(Ctrl) for 24h.

r-u, Western blotting analysis showing the protein levels of NEK7 in total (**r,s**) and mitochondrial (**t,u**) lysates of the HepG2 cells treated with FFA (palmitic acid sodium: oleate sodium, 500 μ M:500 μ M) and the Vehicle (Ctrl, 1%BSA) for 24h.

Data are presented as mean \pm SE. N value indicates independent replicates (from distinct samples). Significant differences are analyzed using unpaired two-tailed *t*-test (**b,d,f,h,j,l,m,o,q,s,u**).

2. Although the authors propose that NEK7 binding stabilizes Complex II conformation, it is paradoxical that NEK7 deficiency increases Complex II activity while destabilizing its structure. This mechanistic disconnect is only partially addressed, and remains speculative without more rigorous structural or in vivo functional data.

Response: Thank the Reviewer for the comments. Sorry for the lack of a more detailed elaboration on this point in the manuscript. In this study, we first observed various abnormalities in hepatocyte mitochondrial function upon NEK7 depletion, including decreased production of ATP, increased membrane potential and mtROS levels along with elevated mitochondrial complex II activity (**Fig.2, Fig.3n**), all of which are the important indicators of reverse electron transfer (RET)¹⁻⁴. Moreover, inhibition of RET significantly alleviated the mitochondrial dysfunction and exacerbated fibrosis induced by NEK7 loss (**Fig.5 and Supplementary Fig.9f**), further supporting the conclusion that loss of NEK7 induces reverse electron transfer (RET) at complex II. Furthermore, through co-immunoprecipitation, mass spectrometry, co-localization staining, and molecular interaction assays, we identified that NEK7 interacts with SDHB, a subunit of mitochondrial complex II (**Fig.3a-d,i-l and Supplementary Fig.4 a-f,i**). However, neither overexpression nor knockdown of NEK7 affected the expression of SDHB and its post-translational modifications, or the assembly of complex II (**Fig.3e-h and Supplementary Fig.4g,h**).

Previous studies have shown that the conformational dynamics of mitochondrial complexes determine their structural stability and activity, which are essential for normal electron transport and energy production⁵⁻⁷. The most stable conformation (forward electron transfer) is not the most reactive in activity, but rather the non-equilibrium conformation (reverse electron transfer) has abnormally high activity⁸. Using molecular dynamics simulations, we demonstrated that the binding of NEK7 to SDHB stabilizes the conformation of mitochondrial complex II (**Fig.3m**). Moreover, overexpression of NEK7

restored the abnormally increased complex II activity and reduced RET-derived ROS (closely associated with high complex activity) (**Supplementary Fig.9g and Fig.3s**). These results provide an explanation for the various mitochondrial abnormalities caused by the loss of NEK7 mentioned above.

3.No human fibrotic tissue samples were analyzed for NEK7 levels. The only human data comes from normal tissue adjacent to hepatoblastoma, which does not reflect fibrosis pathophysiology. As a result, the clinical relevance of the findings is questionable.

Response: Thank the Reviewer for the comments. Sorry for the lack of human fibrotic livers to detect NEK7 levels. We analyzed the RNA-seq data in public GEO dataset (GSE135251) to identify the level of NEK7 in fibrotic livers of NAFLD patients. The NAFLD patients with higher level of liver fibrosis (fibrosis stage:4, n=14) and the control with no liver fibrosis (fibrosis stage:0, n=8) were included for analysis. We found that the level of NEK7 in the fibrotic liver of NAFLD patients showed a decreased trend (with high variability in the data), with a p-value of 0.0909 as analyzed by a two tailed t-test and a p-value of 0.0455 as analyzed by a one tailed t-test (**Supplementary Fig.10m**). More fibrotic liver samples could be included for further analysis to get a better significance. Additionally, we found that the level of NEK7 was significantly decreased in the liver tissues and mitochondria from fibrotic mice induced by CCl₄ or CDAHFD (**Supplementary Fig.10a-I**), which was further confirmed in the HepG2 cells treated with CCl₄ and free fatty acids (**Supplementary Fig.10n-u**). All these results from the mouse fibrosis model and HepG2 cells further support the decreased expression of NEK7 in fibrotic livers.

Overall assessment

While the topic is interesting and the data collection appears technically sound, the above concerns represent major gaps. In particular, the lack of evidence that NEK7 is reduced in fibrosis models or in patients is a critical weakness that undermines the proposed therapeutic rationale. Therefore, I do not support publication of this manuscript in its current form.

Response: We sincerely appreciate the reviewers' valuable and professional comments. Followed the suggestions, additional experiments were conducted

and revisions were made to the manuscript, which have greatly enhanced the quality of this work.

References

- 1 Quinlan, C. L. *et al.* Mitochondrial complex II can generate reactive oxygen species at high rates in both the forward and reverse reactions. *The Journal of biological chemistry* **287**, 27255-27264, doi:10.1074/jbc.M112.374629 (2012).
- 2 Florido, J. *et al.* Melatonin drives apoptosis in head and neck cancer by increasing mitochondrial ROS generated via reverse electron transport. *Journal of pineal research* **73**, e12824, doi:10.1111/jpi.12824 (2022).
- 3 Cadenas, S. Mitochondrial uncoupling, ROS generation and cardioprotection. *Biochimica et biophysica acta. Bioenergetics* **1859**, 940-950, doi:10.1016/j.bbabi.2018.05.019 (2018).
- 4 Mills, E. L. *et al.* Succinate Dehydrogenase Supports Metabolic Repurposing of Mitochondria to Drive Inflammatory Macrophages. *Cell* **167**, 457-470 e413, doi:10.1016/j.cell.2016.08.064 (2016).
- 5 Kampjut, D. & Sazanov, L. A. The coupling mechanism of mammalian respiratory complex I. *Science* **370**, doi:10.1126/science.abc4209 (2020).
- 6 Sazanov, L. A. The mechanism of coupling between electron transfer and proton translocation in respiratory complex I. *J Bioenerg Biomembr* **46**, 247-253, doi:10.1007/s10863-014-9554-z (2014).
- 7 Nakamoto, R. K., Baylis Scanlon, J. A. & Al-Shawi, M. K. The rotary mechanism of the ATP synthase. *Arch Biochem Biophys* **476**, 43-50, doi:10.1016/j.abb.2008.05.004 (2008).
- 8 Hoffman, B. M. *et al.* Differential influence of dynamic processes on forward and reverse electron transfer across a protein-protein interface. *Proceedings of the National Academy of Sciences of the United States of America* **102**, 3564-3569, doi:10.1073/pnas.0408767102 (2005).

Response to Reviewer #4:

From this reviewer's point of view: changes made in the revised manuscript were sufficient to recommend publication.

Response: Very appreciate the Reviewer for the valuable comments which have substantially improved this study.